# Coding of latent variables in sensory, parietal, and frontal cortices during closed-loop virtual navigation

**Jean-Paul Noel[1], Edoardo Balzani[1], Eric Avila[1], Kaushik J Lakshminarasimhan[1,2], Stefania Bruni[1], Panos Alefantis[1], Cristina Savin[1†], Dora E Angelaki[1*†]**

[1]Center for Neural Science, New York University, New York City, United States; [2]Center for Theoretical Neuroscience, Columbia University, New York, United States

**Abstract** We do not understand how neural nodes operate and coordinate within the recurrent action-perception loops that characterize naturalistic self-environment interactions. Here, we record single-unit spiking activity and local field potentials (LFPs) simultaneously from the dorsomedial superior temporal area (MSTd), parietal area 7a, and dorsolateral prefrontal cortex (dlPFC) as monkeys navigate in virtual reality to 'catch fireflies'. This task requires animals to actively sample from a closed-loop virtual environment while concurrently computing continuous latent variables: (i) the distance and angle travelled (i.e., path integration) and (ii) the distance and angle to a memorized firefly location (i.e., a hidden spatial goal). We observed a patterned mixed selectivity, with the prefrontal cortex most prominently coding for latent variables, parietal cortex coding for sensorimotor variables, and MSTd most often coding for eye movements. However, even the traditionally considered sensory area (i.e., MSTd) tracked latent variables, demonstrating path integration and vector coding of hidden spatial goals. Further, global encoding profiles and unit-to-unit coupling (i.e., noise correlations) suggested a functional subnetwork composed by MSTd and dlPFC, and not between these and 7a, as anatomy would suggest. We show that the greater the unit-to-unit coupling between MSTd and dlPFC, the more the animals' gaze position was indicative of the ongoing location of the hidden spatial goal. We suggest this MSTd-dlPFC subnetwork reflects the monkeys' natural and adaptive task strategy wherein they continuously gaze toward the location of the (invisible) target. Together, these results highlight the distributed nature of neural coding during closed action-perception loops and suggest that fine-grain functional subnetworks may be dynamically established to subserve (embodied) task strategies.

**\*For correspondence:**
da93@nyu.edu

[†]These authors contributed equally to this work

**Competing interest:** The authors declare that no competing interests exist.

## Editor's evaluation

This important study investigates distributed neural coding across the three brain areas MST, 7a, and dlPFC in monkeys carrying out a novel behavioural paradigm with a naturalistic closed action-perception-loop developed by the same group previously. The convincing model-based analysis discerns potential influences (e.g. task variables, hidden variables) on firing rates and supports the claim of task-specific sub-networks being formed. The authors provide an important first step to unravel potential drivers of dynamic activity in distributed networks during recurrent action-perception-loops, which should be augmented by future analyses of, for instance, the contribution of changing visual input, especially as the recordings stem from areas involved in processing optical flow, and of signals across different circuit elements like cortical layers.

## Introduction

Despite the closed-loop interplay that exists between action and perception in the real world, our understanding of sensory encoding and the neural architectures supporting goal-directed behavior is largely derived from tasks segregating action from perception, and only sporadically requiring motor output. Arguably, the limited purview of this traditional approach has hindered our ability to understand neural coding for dynamic and flexible behaviors (see, e.g., *Cisek and Kalaska, 2010*; *Gomez-Marin et al., 2014*; *Pitkow and Angelaki, 2017*; *Yoo et al., 2021*, for similar arguments).

First, severing the loop between action and perception disrupts the natural timing that exists between sensory events and internal neural dynamics. In natural vision, for example, eye movements dictate the content, relative resolution, and timing of visual input. Indeed, work from active sensing, for example, *Schroeder et al., 2010*; *Yang et al., 2016*, has shown that neural excitability in primary visual cortex (*Barczak et al., 2019*) and the anterior nuclei of the thalamus (*Leszczynski et al., 2020*) are enhanced at saccade offset – precisely when new observations are made. This enhancement is likely mediated by phase-reset of neural oscillations (*Lakatos et al., 2008*; *Rajkai et al., 2008*) caused by the shifting gaze. In turn, physical movements of the eyes during sensory processing – an aspect absent in most laboratory, binary decision-making tasks – may not only enhance low-level visual encoding but may also favor certain channels of inter-area communication via local field potential (LFP) phase alignment or other coupling mechanisms (e.g., *Jutras et al., 2013*).

Second, the emphasis on tasks with poor dynamics, together with a technology-limited focus on studying one area at a time, has limited our ability to explore how within- and across-area communication enables flexible behavior. This has possibly led to a degeneracy in the number of functions ascribed to each neural node. For instance, notorious redundancy has been observed between parietal and frontal cortices with both areas showing similar properties during visual feature categorization (*Swaminathan and Freedman, 2012*), target selection (*Suzuki and Gottlieb, 2013*), visuo-spatial memory (*Chafee and Goldman-Rakic, 1998*), and working memory (*Olesen et al., 2004*), among others (see *Katsuki and Constantinidis, 2012*, for a review). While this redundancy is certainly an adaptive feature (*Moreno-Bote et al., 2014*; *Driscoll et al., 2017*), the joint characterization of sensory, parietal, and frontal areas during tasks requiring a flow of cognitive operations typical of daily life (e.g., sensing, memorizing, acting) may offer the possibility to study how these regions differ, and how they interact.

To tackle these limitations, we have developed a task requiring observers to catch memorized 'fireflies' in virtual reality (*Lakshminarasimhan et al., 2018*; *Lakshminarasimhan et al., 2020*; *Noel et al., 2020*; *Noel et al., 2021*). This goal-directed virtual navigation task addresses many of the limitations of the traditional approach, while remaining rooted in well-established neural elements: motion detection (*Newsome and Paré, 1988*), optic flow processing (*Britten, 2008*), and navigation (*Ekstrom et al., 2018*). Animals first detect a briefly flashed target, much like the blinking of a firefly. Then, they use a joystick controlling linear and angular velocity to navigate via path integration to the memorized location of this target (*Figure 1*). Importantly, the observers' eyes are free to sample from their sensory surroundings, and trials last on the order of 2–3 s. In turn, this task requires integration of sensory evidence over protracted time periods, allows for active sensing, and engages the real-world loop between observations, beliefs, actions, and environmental states (*Figure 1A*). Importantly, the critical task variables are latent. Namely, observers must dynamically estimate their self-location (by accumulating velocity flow vectors) and that of the target (i.e., a hidden spatial goal). This dynamic computation of latent variables may offer a window into the mechanisms of internal computation (e.g., see *Lakshminarasimhan et al., 2020*).

Here, we leverage this 'firefly task' to simultaneously characterize the encoding profiles of single units in sensory (dorsomedial superior temporal area, MSTd), parietal (area 7a), and frontal (dorsolateral prefrontal cortex, dlPFC) areas during closed-loop goal-directed behavior (*Figure 2—figure supplement 1*). We record from these regions (MSTd, 7a, and dlPFC) because they form a series of reciprocally interconnected areas (*Andersen et al., 1990*; *Rozzi et al., 2006*) that are well established in the processing of optic flow for self-motion, sensorimotor transformations, and belief formation/ working memory, all critical elements of the task (see, e.g., *Kravitz et al., 2011*; *Constantinidis and Klingberg, 2016*; *Christophel et al., 2017*; *Noel and Angelaki, 2022*, for reviews). We observe that all areas probed – including MSTd, a classically considered sensory area – encode latent task variables. Further, global encoding profiles and unit-to-unit couplings suggested a functional subnetwork

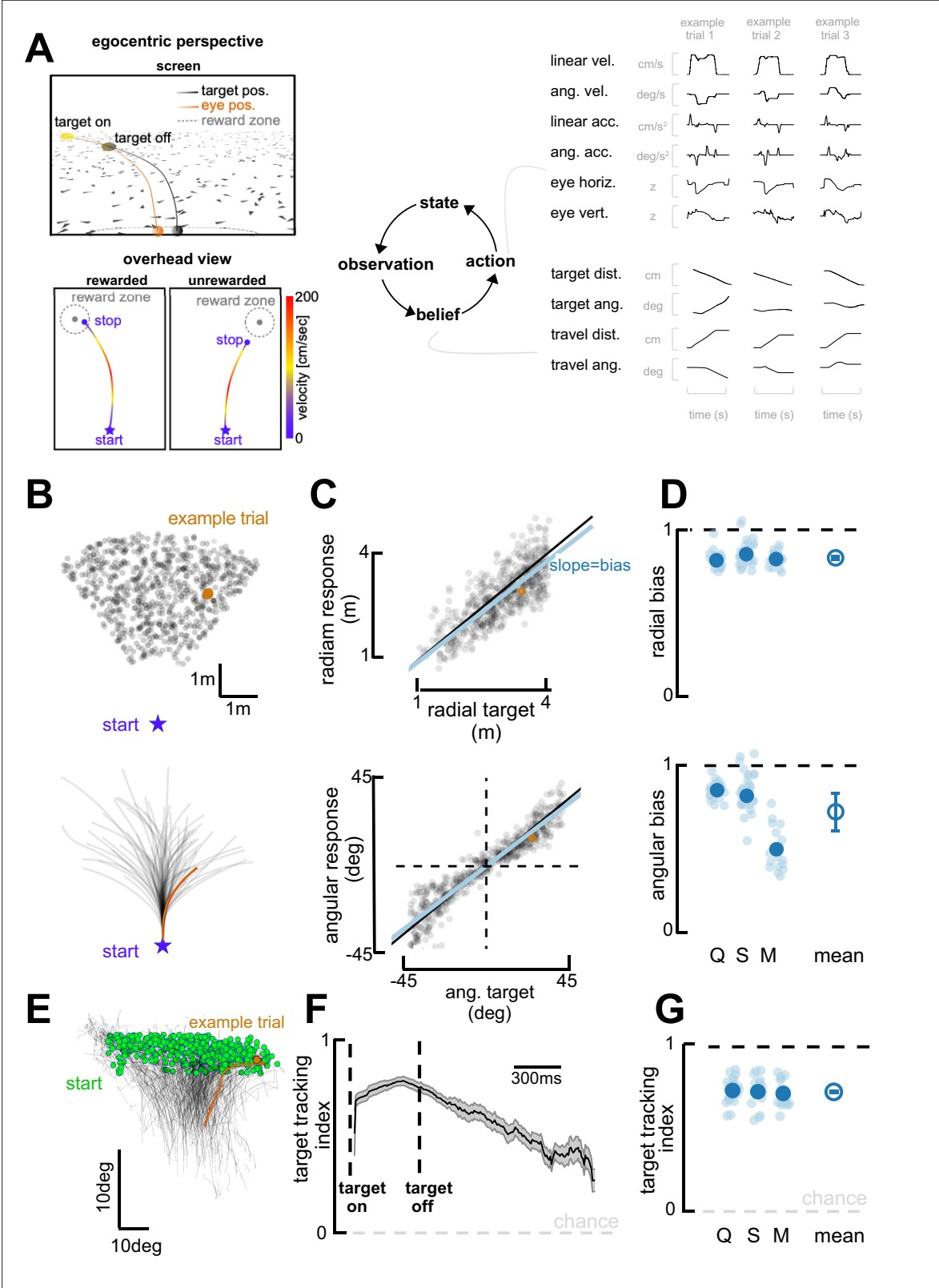

**Figure 1.** Task and behavioral results. (**A**) Behavioral task. Top left: Egocentric perspective of the monkey during the firefly task, emphasizing the ground plane optic flow elements (flashing triangles), the fact that the target disappears (target off), and eye positions (following target position perfectly when the target is on, and then continuing on the correct path but deviating with time). Bottom left: Overhead view of the monkey, starting location (star), accelerating and progressively re-orienting to face the firefly, before de-accelerating and stopping at (left; rewarded) or near (right: unrewarded) the

*Figure 1 continued on next page*

*Figure 1 continued*

location of the now invisible target. Right: This task involves making observation of the sensory environment (composed of optic flow), using these observations to generate a dynamic belief (of the relative distance to target), and producing motor commands based on the current belief, which in turn updates the state of the environment. Right: Continuous variables are shown for three example trials. (**B**) Top: spatial distribution of target positions across trials. Bottom: monkey trajectories. The orange dot and trajectory are an example trial, maintained throughout the figure. (**C**) Example session. Radial (top) and angular (bottom) endpoint (y-axis) as a function of target (x-axis). Gray dots are individual trials, black line is unity, and blue line is the regression between response and target (slope of 1 indicates no bias). (**D**) Radial (top) and angular (bottom) bias (slope,=1 means no bias, >1 means overshooting, and <1 means undershooting) for every session (transparent blue circles) and monkey (x-axis, dark blue circles are average for each monkey, Q, S, and M). Rightmost circle is the average across monkeys and error bars represent ±1 SEM. Overall, monkeys are fairly accurate but undershoot targets, both radially and in eccentricity. (**E**) Eye trajectories (green = eye position at firefly offset) for an example session, projected onto a two-dimensional 'screen'. Eyes start in the upper field and gradually converge in the lower center (where the firefly ought to be when they stop). (**F**) Target-tracking index (variance in eye position explained by prediction of fixating on firefly) for an example session as a function of time since firefly onset and offset. (**G**) Average target-tracking index within 1 s for all sessions (light blue) and monkeys (dark blue) showing the monkeys attempt to track the invisible target.

composed by dlPFC and MSTd (and not area 7a, as would be predicted from anatomy). We suggest this MSTd-dlPFC subnetwork reflects the natural task strategy wherein monkeys continuously gaze toward the location of the (invisible) target.

## Results

### Monkeys navigate to remembered targets employing natural task strategies

Targets ('fireflies') were displayed for 300 ms on a ground plane composed of triangular optic flow elements that appeared transiently (250 ms) with random orientations. The density of these optic flow elements was 5 elements/$m^2$. A single target was presented per trial. The firefly targets were uniformly distributed (across trials) within a radial range of 1–4 m, and an angular eccentricity spanning from –40° (leftward) to 40° (rightward) of visual angle (*Figure 1A and B*). Performance feedback was provided at the end of each trial in the form of juice reward for correctly stopping within a reward boundary (see *Methods* for further details).

Visualizing example trials shows that monkeys followed curvilinear trajectories and stopped near the latent targets (*Figure 1B*). To quantify their performance, we expressed the monkeys' trajectory endpoints and firefly locations in polar coordinates, with an eccentricity from straight-ahead ($\theta$) and a radial distance ($r$). *Figure 1C* shows radial (top; $r$ vs. $\tilde{r}$ slope = 0.90; $R^2$=0.55) and angular (bottom; $\theta$ vs. $\tilde{\theta}$ slope = 0.95; $R^2$=0.78) responses as a function of target location for an example session (omitting 18% of trials the animals aborted early or never stopped, see *Methods*). As shown by this example session, and confirmed across three animals and all sessions (Monkey Q, 27 sessions; S, 38 sessions; M, 17 sessions; *Figure 1D*), monkeys were on average accurate, but tended to undershoot targets ($r$ vs. $\tilde{r}$ slope, mean = 0.892, 95% CI=[0.860, 0.914]; $\theta$ vs. $\tilde{\theta}$ slope, M=0.792, 95% CI=[0.762, 0.842]). Undershooting targets is, in fact, the optimal strategy given that the uncertainty over one's own and the firefly's position scales supra-linearly with time and distance travelled (see *Lakshminarasimhan et al., 2018*).

Most interestingly, this task affords observers with degrees of freedom that may reveal their innate task strategy. For instance, they are not required to fixate and instead we may use their eye movements as an indicator of their internal beliefs (*Lakshminarasimhan et al., 2020*). Indeed, over the course of individual trajectories, the monkeys' eyes move downward and become less eccentric (*Figure 1E*), as if tracking the (invisible) target progressively becoming nearer and aligned with straight ahead (*Lakshminarasimhan et al., 2020*; also see *Ilg and Thier, 2003*; *Ilg and Thier, 1999*). This behavior was quantified by deriving predictions for the binocular position of the observer's eyes, assuming the monkeys maintained fixation at the center of the target throughout the trial. Then, we expressed a target-tracking index as the square root of the fraction of variance in the observed eye position that was explained by this prediction (see *Lakshminarasimhan et al., 2020*, for further details). An index of 1 implies that the subject consistently looked at the center of the firefly, while a value of 0 denotes a lack of correspondence between target and gaze locations. The target-tracking index was high at firefly onset, highest at firefly offset, and remained above chance throughout the course of the trial

(*Figure 1F* shows an example session). Across all animals and sessions, the target-tracking index averaged 0.73 (95% CI = [0.72, 0.74]) during the first second of each trial. This suggest that, while not a task requirement, animals developed the strategy of fixating on the visible firefly and then attempted to maintain their gaze on it even after the target had disappeared (*Figure 1G*, see *Lakshminarasimhan et al., 2020*, for a full description of this phenomenon and demonstration that pursuing the invisible firefly with our eyes results in improved task performance).

Altogether, the animals accumulated optic flow velocity signals (*Alefantis et al., 2021*) into an evolving (and largely accurate) estimate of self-position. They used this estimate to path integrate to the location of latent targets over prolonged periods of time. In addition, they seemingly sampled from their environment in an intelligent manner, tracking the (invisible) target with their eyes in what is seemingly an embodied mnemonic strategy.

## Patterned mixed selectivity across sensory, parietal, and frontal cortices

We recorded single-unit activity from a large number of neurons in MSTd (231 units), area 7a (3200 units), and dlPFC (823 units) across 82 sessions while monkeys performed the firefly task (*Figure 2A* and *Figure 2—figure supplement 1*).

To quantitatively account for the encoding properties of neurons within this task – wherein no two trials are alike (see *Figure 2B* for rasters demonstrating heterogeneity in spike times and trial durations) – we fit spike trains to a Poisson generalized additive model (P-GAM). The P-GAM we leveraged was purposely developed to efficiently and robustly estimate encoding profiles during closed-loop and naturalistic tasks such as the 'firefly task' (*Balzani et al., 2020b*). Namely, beyond capturing arbitrary non-linearities and handling collinear predictors (*Dormann et al., 2013*), this encoding model has the strong advantage of inferring marginal confidence bounds for the contribution of each feature to neural responses (see *Balzani et al., 2020b*, for details). This property allows us to identify the minimal set of task variables that each neuron is responsive to (setting statistical significance to p<0.01), while circumventing the need for computationally demanding (and often unstable) model selection procedures – particularly given the large number and time-varying nature of the variables in this task. Indeed, beyond the numerical results in *Balzani et al., 2020b*, demonstrating the ability to recover the ground-truth in artificial data, here we show that the P-GAM will capture the simplest possible interpretation of the neural responses in the statistical regime of our data, that is, even when neurons are weakly firing and predictors are correlated (see *Figure 2—figure supplement 2*).

In addition to continuous sensorimotor (e.g., linear and angular velocity and acceleration) and latent variables (e.g., distance from origin and to target, *Figure 1A*), as well as discrete task events (e.g., time of target onset, as well as movement onset and offset), we included elements of brain dynamics in the encoding model. These internal dynamics are most often not considered in accounting for task-relevant neural responses, yet they fundamentally shape spiking activity. These latter variables included the phase of LFP in different frequency bands (theta: 4–8 Hz; alpha: 8–12 Hz; beta: 12–30 Hz), and causal unit-to-unit coupling filters within (i.e., spike history, 36 ms wide temporal filter) and between units, both within (36 ms wide temporal filter) and across cortical areas (600 ms wide temporal filters, *Figure 2C*, see *Methods*). These coupling filters capture noise correlations, that is, the signal-independent relationship between one neuron firing and the likelihood that this same (i.e., spike history) or another neuron (i.e., coupling filter) will fire at a given time delay (see *Hart and Huk, 2020*). In total, the encoding model had 17 (analog or digital) task inputs (*Figure 2C*), in addition to hundreds of potential lateral and across region coupling filters (see *Figure 2—figure supplement 2B* for the distribution of the number parameters in the full and reduced encoding models, and *Figure 2—figure supplement 2C* for demonstration that while the reduced model had a fifth of the parameters present in the full model, its cross-validated ability to account for observed spiking was unchanged). The fitted model accounts well for observed spiking activity (see raw and model reconstructed spiking activity of simultaneously recorded units in *Figure 2D*, mean pseudo-$R^2$=0.072).

*Figure 2E* shows the empirical (black) and P-GAM reconstructed (colored) firing rate to task variables in an example MSTd, 7a, and dlPFC neuron. The striking feature of these examples is that all units responded to, or were modulated by, a multitude of task variables. For instance, each of these example neurons responded at the time of firefly onset. The 7a neuron showed a strong dependency not only to sensorimotor variables, but also to LFP phases (in all frequency ranges). Finally,

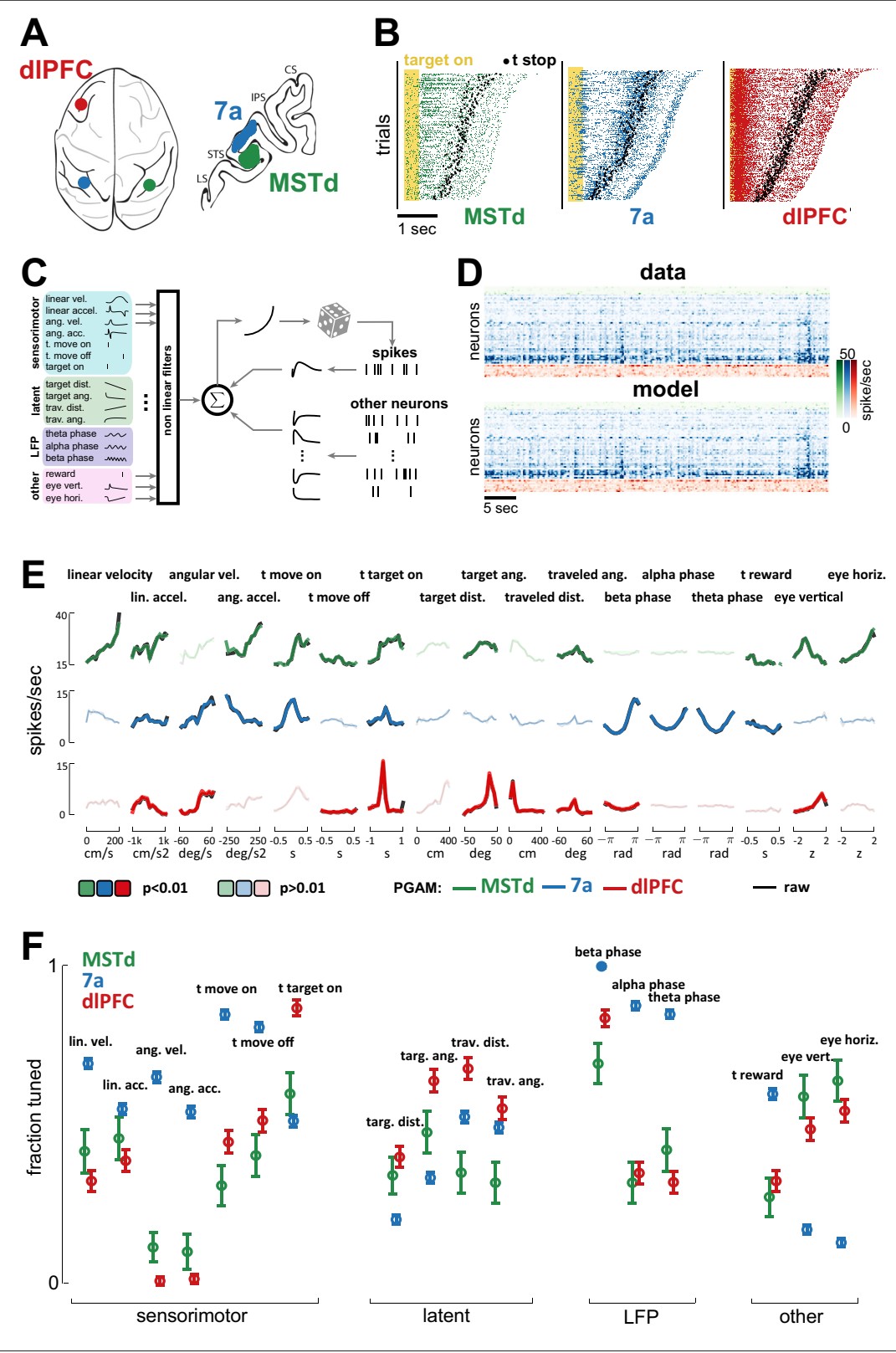

**Figure 2.** Dorsomedial superior temporal area (MSTd), area 7a, and dorsolateral prefrontal cortex (dlPFC) encode a heterogeneous mixture of task variables. (**A**) Schematic of brain areas recorded. (**B**) Raster plots of spiking activity in MSTd (green), 7a (blue), and dlPFC (red). Shaded yellow area represents the time of the target being visible, and black dots represent the timing of movement offset. Trials are sorted from shortest (bottom) to longest (top)

*Figure 2 continued on next page*

*Figure 2 continued*

duration of the trial. (**C**) Schematic of the Poisson generalized additive model (P-GAM) used to fit spike trains. The task and neural variables used as input to the model were: linear and angular velocity and acceleration, time of movement onset and offset, time of firefly onset, distance and angle from origin and to target, time of reward, vertical and horizontal position of the eyes, and ongoing phase of LFP at theta, alpha, and beta bands. (**D**) Top: random snippet of spiking activity of simultaneously recorded neurons (green = MSTd; blue = 7a; red = dlPFC). Bottom: corresponding cross-validated prediction reconstructed from the P-GAM. The average cross-validated pseudo-$R^2$ was 0.072 (see *Colin Cameron and Windmeijer, 1997*). (**E**) Responses from an example MSTd, 7a, and dlPFC neuron (black), aligned to temporal task variables (e.g., time of movement onset and offset), or binned according to their value in a continuous task variable (e.g., linear velocity). Colored (respectively green, blue, and red for MSTd, 7a, and dlPFC) lines are the reconstruction from the reduced P-GAM. The responses are opaque ($p<0.01$) or transparent ($p>0.01$), according to whether the P-GAM kernel for the specific task variable is deemed to significantly contribute to the neuron's firing rate. (**F**) Fraction of neurons tuned to the given task variable, according to brain area. Error bars are 99% CIs, and thus non-overlapping bars indicate a pair-wise significant difference.

The online version of this article includes the following figure supplement(s) for figure 2:

**Figure supplement 1.** Recoding sites.

**Figure supplement 2.** Poisson generalized additive model (P-GAM) controls.

**Figure supplement 3.** Effect sizes in the firing rate space for neurons deemed to code for sensorimotor variables.

**Figure supplement 4.** Effect sizes in the firing rate space for neurons deemed to code for latent variables.

**Figure supplement 5.** Effect sizes in the firing rate space for neurons deemed to phase lock to local field potential (LFP).

**Figure supplement 6.** Effect sizes in the firing rate space for neurons deemed to code for reward and eye position.

**Figure supplement 7.** Speed and direction discrimination index for neurons in dorsomedial superior temporal area (MSTd), 7a, and dorsolateral prefrontal cortex (dlPFC).

**Figure supplement 8.** Illustration of spike-local field potential (LFP) phase locking.

**Figure supplement 9.** Pairwise phase consistency.

**Figure supplement 10.** Stability in the fraction of neurons tuned to different task variables.

**Figure supplement 11.** Task engagement drives neural tuning.

and perhaps most surprisingly, the MSTd neuron reflected not only sensorimotor and eye position variables, but also latent ones, such as the angular distance to the (invisible) target. We used the P-GAM to factorize the contribution of different elements of the task to a neuron's overall firing rate (i.e., perform credit assignment). The example neurons in *Figure 2E* are opaque (contributes) or transparent (does not contribute) for different task variables, according to the P-GAMs estimate of their factorized contribution. Thus, there may be cases where there are clear evoked responses (e.g., time of movement onset in the dlPFC example neuron, *Figure 2E*, bottom) yet the P-GAM estimated the neuron to not be tuned to this variable, likely due to the correlated nature of input statistics during this naturalistic task (i.e., this neuron appears tuned to linear acceleration, which correlates with the time of movement onset). In *Figure 2—figure supplements 3–6* we quantify effect sizes by comparing the modulation in firing rates, as well as the mutual information between these and a given variable, for the population of neurons significantly coding or not for a given task variable. Similarly, to provide a point of comparison with prior work studying optic flow processing, in *Figure 2—figure supplement 7*, we quantify the speed (i.e., liner velocity) and direction (i.e., angular velocity) discrimination index (see *Methods* and e.g., *Chen et al., 2008*; *Avila et al., 2019*) for neurons in MSTd, 7a, and dlPFC.

The fraction of neurons tuned to different task variables demonstrated a patterned mixed selectivity (*Figure 2F*). Namely, the fraction of neurons tuned to sensorimotor variables was highest in area 7a (linear and angular velocity, linear and angular acceleration, time of movement onset and offset, all $p<0.01$; Cohen's $d$ linear acceleration $d=0.2$, all the rest $d>0.76$), while most neurons in MSTd coded for eye position (58.8% and 63.3% respectively for vertical and horizontal eye position). Interestingly, a large fraction of dlPFC neurons also coded for eye position (dlPFC vs. 7a, vertical and horizontal eye position, $p<6.4 \times 10^{-17}$, all Cohen's $d>3$), putatively reflecting the fact that the gaze indexes the internal belief over firefly position (*Lakshminarasimhan et al., 2020*). Neurons in area 7a showed a

strong dependency to the phase of ongoing LFP fluctuations (see *Figure 2—figure supplement 8* for further illustrations of this effect as quantified by the P-GAM, and *Figure 2—figure supplement 9* for corroborative evidence by pairwise phase consistency, *Vinck et al., 2010*), a fact that was less observed in dlPFC or MSTd (all p<7.1 × 10⁻⁸, all $d$>1.02). Lastly, and most relevant to the task of path integrating to the location of invisible fireflies, we observed that the greatest fraction of neurons encoding for both path integration (i.e., travelled distance and angle turned) and the distance and angle to the target (i.e., spatial goal) were in dlPFC (ranging from 30.6% to 57.5%). Interestingly, while 7a had more neurons coding for path integration than MSTd (all p<1.6 × 10⁻⁵, all Cohen's $d$>0.54), the latter area had a greater fraction of neurons coding for the latent distance and angle to target than 7a did (all p<6.1 × 10⁻⁵, all Cohen's $d$>0.41). In the supplement we demonstrate that this coding was stable (contrasting odd vs. even trials; *Figure 2—figure supplement 10*) and task-relevant (*Figure 2—figure supplement 11*), in that passive viewing of the same stimuli did not elicit a comparable fraction of neurons tuned to task variables in 7a (passive viewing data in MSTd and dlPFC were unavailable). The fraction of neurons aligned with the phase of LFP in different frequency bands remained stable across passive and active viewing conditions, particularly in the beta band (all frequencies, active vs. passive, p=0.13; beta band, p=0.51). Altogether, the encoding pattern across areas may suggest that while dlPFC is critically involved in estimating the relative distance between self and target, 7a may be preferentially involved in the process of path integration, while somewhat unexpectedly, MSTd may play an important role in keeping track of the latent spatial goal.

## Cortical codes for path integration and vector coding of spatial goals

Beyond the frequency with which we observe neurons tuned to the angle and distance from the origin (i.e., path integration) and to the target (i.e., vector coding of spatial goals), we may expect the distributions of preferred distances and angles to also be informative. Of note, distance/angle from origin and to the target are not the reciprocal of one another given that the target location varies on a trial-by-trial fashion. In other words, the travelled distance and the distance to target may correlate within a trial (but need not, given under- vs. overshooting) but certainly do not across trials (e.g., a distance of, say, 100 cm from the origin could corresponding to a whole host of distances from target). In *Figure 3A* we show rasters of representative neurons tuned to the distance from origin (example neuron 1) and to target (example neuron 2). The neuron tuned to the distance to target (example neurons 2) is not tuned to a particular distance from origin, but does demonstrate a patterned firing rate, discharging at further distances as the animal travels further. The third example (*Figure 3A*) is tuned to movement stopping, and demonstrates a pattern similar to the neuron tuned to distance to target when plotted as a function of distance from origin (*Figure 3A*, third vs. fifth panel), but not when visualized as a function of distance to target. The distributions shown in *Figure 3B* illustrate that the preferred distances/angles from origin and to target spanned the entire range of angles and distances visited, demonstrating a full basis set covering the feature space.

Across all animals, angles specifying a 0° offset from the heading at origin were over-represented (*Figure 3C*, top row, proportions across all animals), while the 0° offset from target location was under-represented (*Figure 3C*, second row). Instead, particularly in area 7a and dlPFC, the distribution of preferred angles to target was bimodal (bimodality coefficients, null = 0.55, MST = 0.57, 7a=0.89, dlPFC = 0.80, 7a and dlPFC p<0.05), with many neurons showing a preference for ~45–90° away from target, either leftward or rightward. This pattern is in stark contrast with observations from the bat's hippocampus, where vector coding of hidden spatial goals exists and a goal angle of 0° is over-represented (*Sarel et al., 2017*). Speculatively, this reversed pattern of preferences between cortex (here) and hippocampus (*Sarel et al., 2017*) may suggest that while the former plays a role in guiding heading toward the desired state (akin to an error or correction signal), the hippocampus maintains this heading.

In terms of radial distance, MSTd, area 7a, and dlPFC all showed an over-representation of units coding for distances near to, as opposed to far from, the starting location (*Figure 3C*, third row, <~150 cm). On the other hand, we observed a clear differentiation between cortical areas regarding their preferred distance to target. Area 7a showed a strong preference for nearby targets with approximately 50% of units coding for targets within ~30 cm. Neurons in MSTd and dlPFC, on the other hand, responded primarily at intermediary and far distances from the target (~200–400 cm from target; *Figure 3C*, fourth row). The preference for distances further from the target in MSTd

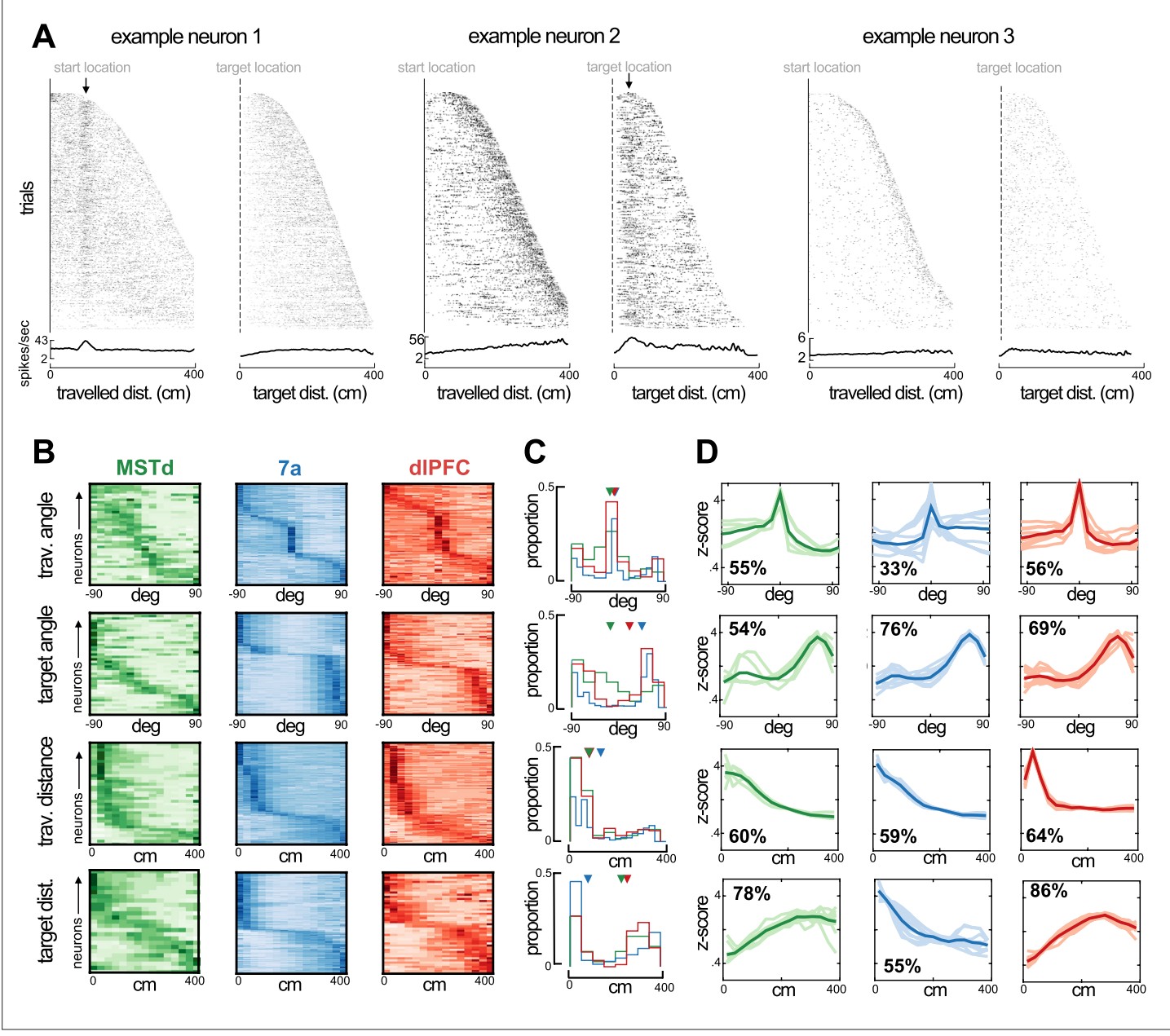

**Figure 3.** Preferred angle and distance from origin and to target in dorsomedial superior temporal area (MSTd), 7a, and dorsolateral prefrontal cortex (dlPFC). (**A**) Rasters and average firing rate of three example neurons, sorted by their maximal distance from origin and to target. The first example neuron (left) responds at a distance of ~100 cm from origin and is not modulated by distance to target. The second example (middle) responds to a close distance to target (~30 cm). Arrows at the top of these rasters indicate the preferred distance from origin (example 1) and to target (example 2). We include a third example (tuned to movement stop) as a control, demonstrating that responding to a distance near the target and to stopping behavior are distinguishable. (**B**) Heatmaps showing neural responses (y-axis) sorted by preferred angles from origin (top), angle to target (second row), distance from origin (third row), and distance to target (bottom row) for MSTd (green), 7a (blue) and dlPFC (red) in Monkey S (data simultaneously recorded). Darker color indicates higher normalized firing rate. Neurons were sorted based on their preferred distances/angles in even trials and their responses during odd trials is shown (i.e., sorting is cross-validated, see Methods). (**C**) Histograms showing the probability of observing a given preferred angle or distance across all three monkeys. Inverted triangles at the top of each subplot indicate the median. Of note, however, medians may not appropriately summarize distributions in the case of bimodality. (**D**) We clustered the kernels driving the response to angle or distance to origin and from the target. Here, we show 10 representatives from each cluster (thin and transparent lines), as well as the mean of the cluster as a whole (ticker line). The inset quantifies the percentage of tunings within a particular area and for the particular variable that were deemed to belong within the cluster depicted (the most frequent one).

The online version of this article includes the following figure supplement(s) for figure 3:

*Figure 3 continued on next page*

*Figure 3 continued*

**Figure supplement 1.** Latent tuning functions (radial and angular distance to target and from origin) encountered in dorsomedial superior temporal area (MSTd) (green), area 7a (blue), and dorsolateral prefrontal cortex (dlPFC) (red).

**Figure supplement 2.** Sensorimotor tuning functions to continuous variables (linear and angular velocity and acceleration) encountered in dorsomedial superior temporal area (MSTd) (green), area 7a (blue), and dorsolateral prefrontal cortex (dlPFC) (red).

**Figure supplement 3.** Temporal kernels (time of movement onset and offset, time of target onset and reward presentation) encountered in dorsomedial superior temporal area (MSTd) (green), area 7a (blue), and dorsolateral prefrontal cortex (dlPFC) (red).

**Figure supplement 4.** Tuning functions to eye position variables (vertical and horizontal) encountered in dorsomedial superior temporal area (MSTd) (green), area 7a (blue), and dorsolateral prefrontal cortex (dlPFC) (red).

and dlPFC seemingly concords with their frequent tuning to eye position, and the fact that the eyes attempt to pursue the hidden target (i.e., as if these areas were involved in planning and sampling from a distance).

Lastly, to depict the full shape of the kernels encoding the distance and angle from the origin and to the target, we performed statistical clustering of these kernels, separately for each area and task variable. *Figure 3D* shows the mean and 10 example tuning functions for the most frequently present cluster within each area. The most noticeable difference between areas is that while MSTd and dlPFC prefer distances far from the target, area 7a responds mostly to distances near it (*Figure 3C*, bottom row). In *Figure 3—figure supplements 1–4* we depict all the different types (i.e., high-dimensional clusters) of tuning functions that exist for all the different task variables (e.g., linear and angular velocity, horizontal and vertical eye position, etc.) in MSTd, area 7a, and dlPFC.

## Single-cell properties and unit-to-unit coupling suggest two distinct functional subnetworks

Both the fraction of neurons tuned to different task variables (*Figure 2*) and the distribution of preferred angles and distances from origin and to target (*Figure 3*) show a surprising degree of coding similarity between MSTd and dlPFC. In turn, to systematically examine how the encoding of all task variables, and not solely a select few, varied across cortical areas, we employed a statistical clustering approach (see, e.g., *Minderer et al., 2019*, for a similar approach).

First, we leveraged the knowledge of which variables each neuron was significantly tuned to (e.g., *Figure 2F*), and attempted clustering neurons based on this binary data, each neuron being defined by a size 17 binary vector (i.e., tuned or not tuned) corresponding to the 17 task variables defined in *Figure 2*. This approach (see *Methods* for details) showed that nearly all MSTd (89%) and dlPFC (94%) neurons were categorized as belonging within the same cluster (*Figure 4A*, cluster number 1), one that was defined by true mixed selectivity (*Figure 4B*, top left). In contrast, area 7a neurons appeared in three distinct clusters (*Figure 4A*, cluster numbers 1–3, respectively, 36%, 22%, and 31%). Cluster 2 was characterized by a strong selectivity for sensorimotor variables and firing in alignment with specific LFP phases (*Figure 4B*, top center), while Cluster 3 was characterized by a near complete lack of tuning to latent variables and eye position (*Figure 4B*, top right). Other cluster types existed, for instance composed of neurons selectively tuned to the ongoing phase in LFP bands but no other task variable (*Figure 4A and B*, Cluster 4), or driven also by motor onset and offset (*Figure 4A and B*, Cluster 5). These remaining clusters were, however, less common (~1–5%). This analysis was conducted with the full dataset (4254 neurons in total), yet in the supplement (*Figure 4—figure supplement 1A*) we confirm that the results are unchanged when subsampling from areas with more neurons (7a and dlPFC) to match the number present in MSTd (231 neurons). Together, this pattern of clustering results based on whether neurons were tuned to different task variables demonstrated a surprising degree of coding similarity between MSTd and dlPFC, which are in turn different from area 7a.

Subsequently, we questioned whether utilizing knowledge of the shape of these tuning functions (as opposed to simply whether tuning to a variable was significant or not) would dissociate between neurons in each area. Tuning functions for all task variables of a given neuron were stacked, then all neurons were projected onto a low-dimensional manifold via Uniform Manifold Approximation and Projection (UMAP; *McInnes et al., 2020*), and clustered on this projection via DBSCAN (Density-Based Spatial Clustering of Applications with Noise; *Ester, 1996*). Area 7a and dlPFC neatly segregated from one another, while neurons from MSTd could be found along a continuum from area 7a to

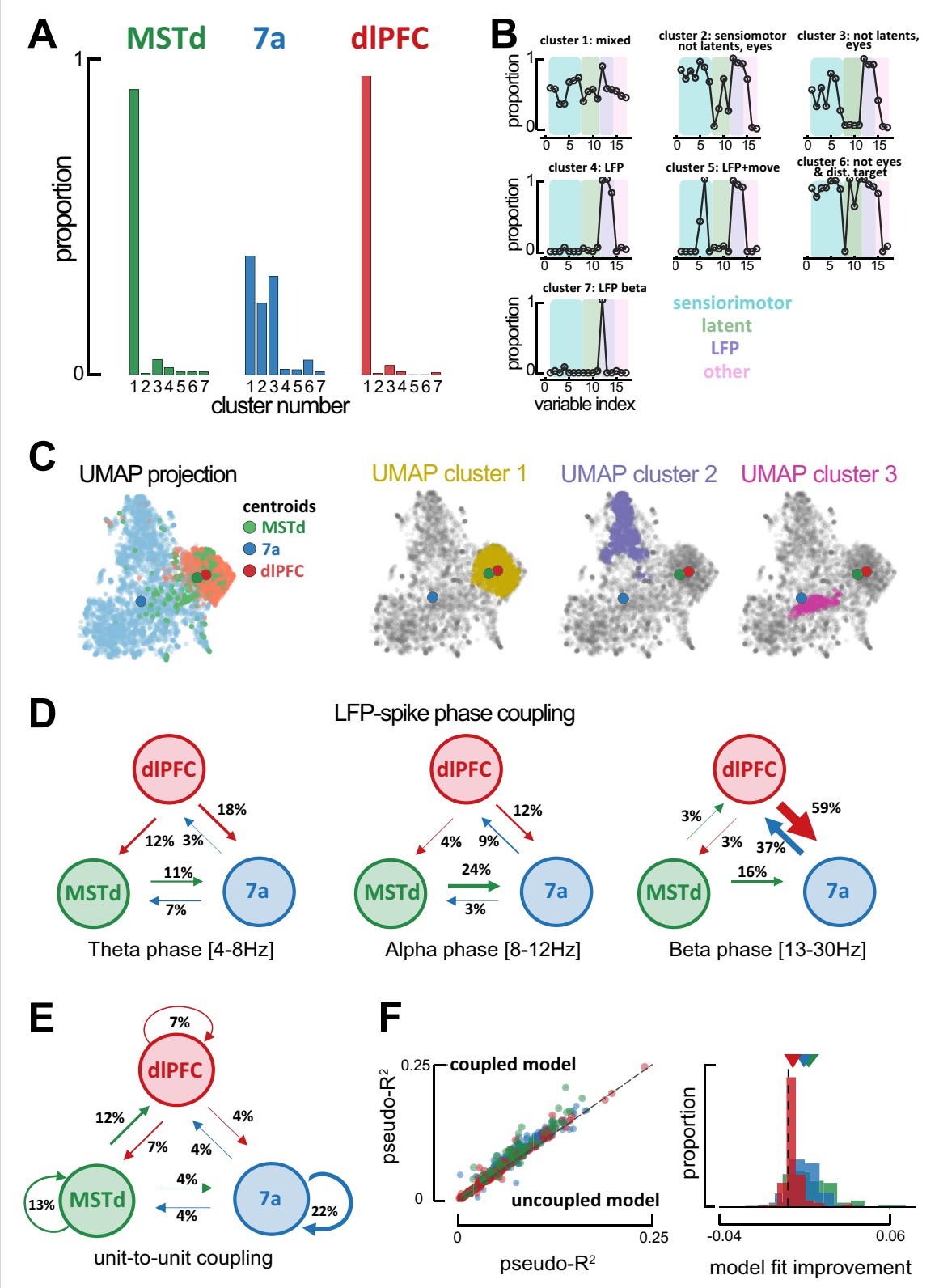

**Figure 4.** Global single-unit encoding profiles and unit-to-unit coupling properties suggest a common functional role for dorsomedial superior temporal area (MSTd) and dorsolateral prefrontal cortex (dlPFC). (**A**) Proportion of neurons being classified into distinct cluster (x-axis, seven total) according to which task parameters they were significantly tuned to. (**B**) Fraction of neurons tuned to each of the 17 task variables (order is the same as in *Figure 2C, E, F*) according to cluster. (**C**) Uniform Manifold Approximation and Projection (UMAP) of the tuning function shapes, color coded by

*Figure 4 continued on next page*

*Figure 4 continued*

brain area (first column) or Density-Based Spatial Clustering of Applications with Noise (DBSCAN) cluster (second, third, and forth column). (**D**) Fraction of neurons whose spiking activity is phase-locked to local field potential (LFP) phases in other areas, in theta (leftmost), alpha (center) and beta (rightmost) bands. An arrow projecting from, for example, MSTd to 7a (center, 0.24), indicates that the neuron in area 7a is influenced by ongoing LFP phase in MSTd. Width of arrows and associated number indicate the proportion of neurons showing the particular property. (**E**) As (**D**) but for unit-to-unit coupling. An arrow projecting from, for example, MSTd to dlPFC indicates that the firing of a neuron in MSTd will subsequently influence spiking activity in dlPFC. (**F**) Left: Cross-validated pseudo-$R^2$ of the full encoding model (y-axis) and a reduced model without within and across area unit-to-unit coupling (x-axis). Right: Probability distribution of the change in pseudo-$R^2$ from the reduced to the full model.

The online version of this article includes the following figure supplement(s) for figure 4:

**Figure supplement 1.** Clustering results, subsampling from neurons in dorsolateral prefrontal cortex (dlPFC) and 7a to match the number of units recorded from in dorsomedial superior temporal area (MSTd).

**Figure supplement 2.** Uniform Manifold Approximation and Projection (UMAP) (**McInnes et al., 2020**) space color coded by mutual information with each task variable.

**Figure supplement 3.** Characteristics of coupling functions within and across areas.

**Figure supplement 4.** Fraction of neurons coupled in 7a and dorsolateral prefrontal cortex (dlPFC) as a function of probe (Utah array or linear probe).

---

dlPFC. Notably, however, the centroid of MSTd was 6.49 times closer to the centroid of dlPFC than area 7a (*Figure 4C*, top row. Note that *Becht et al., 2018*, have shown UMAP to conserve global structure and thus allows for a meaningful interpretation of distances). This finding also holds when subsampling from 7a and dlPFC to match the number of units present in MSTd (100 iterations, MSTd-dlPFC distance was 5.56 times closer than MSTd-7a, 95% CI = [4.07, 7.33]; *Figure 4—figure supplement 1B*). DBSCAN clustering highlighted three main clusters within this low-dimensional projection of tuning functions. Cluster 1 (*Figure 4C*, second row) contained 81.5% of all dlPFC neurons, 58.9% of all MSTd neurons, and 3.6% of all neurons from 7a. Clusters 2 and 3 exclusively contained neurons from area 7a (*Figure 4C*, third and fourth row). Thus, just as the clustering based on what variables were neurons tuned to, clustering based on tuning shapes also highlighted a stronger similarity in the encoding properties of MSTd and dlPFC, as opposed to either of the former and area 7a (see *Figure 4—figure supplement 2* for UMAPs color coded by the mutual information between neural responses and a particular task variable).

Given the similarity in encoding profiles between MSTd and dlPFC, we next examined inter-area global synchronization and unit-to-unit coupling to question whether these two areas formed an interacting functional subnetwork. We examined inter-area coordination both from the standpoint of coarse grain LFP-to-spike phase locking, and via finer resolution unit-to-unit couplings as estimated from the P-GAM (*Balzani et al., 2020b*).

Spiking activity in some units was well explained by the ongoing LFP phase in different regions (above and beyond the ongoing LFP phase in their local area, see *Figure 2F* and *Figure 2—figure supplements 8 and 9*). Most notably, 37% of neurons in dlPFC were tuned to the ongoing LFP phase within the beta range in area 7a. An even greater proportion of units showed the reciprocal property, with 59% of neurons in area 7a being modulated by the ongoing phase of beta-band LFP in dlPFC. A considerable proportion of 7a units were also modulated by the ongoing phase of alpha-band LFP in MSTd (24%, *Figure 4D*). Globally, therefore, phase locking of spiking activity to LFP phase in different areas reflected known anatomical connections, with reciprocal projections between MSTd and 7a, as well as between 7a and dlPFC (*Andersen et al., 1990*; *Rozzi et al., 2006*). Interestingly, the putative 'feedback' influence from dlPFC to area 7a (potentially reflecting the 'belief' guiding motor action) was stronger than the putative 'feedforward' influence of 7a onto dlPFC. The lowest frequency we indexed (theta, 4–8 Hz) seemed to be reserved for putative 'feedback' influence of dlPFC onto both MSTd (12%) and area 7a (18%).

Finer grain unit-to-unit coupling was sparse, and within each area the probability of two units being functionally connected decreased as a function of the distance between neurons (*Figure 4—figure supplement 3A*). The overall likelihood of two units being coupled within a given area changed as a function of brain area and was modulated by task engagement (active vs. passive viewing in area 7a; *Figure 2—figure supplement 11C*), but not as a function of probe type used (Utah array or linear probe, see *Figure 4—figure supplement 4*). There were more units coupled in MSTd (13%, corrected for distance between electrodes, see *Methods*) and area 7a (22%) than in dlPFC (7%, *Figure 4E*), potentially reflecting an increase in the dimensionality of the neural code (i.e., decrease

in redundancy) as we move from sensorimotor to cognitive areas. More importantly, the across area unit-to-unit coupling did not reflect global anatomical trends, and instead supported the functional association between MSTd and dlPFC, as suggested by the encoding profiles of these areas. Twelve percent of neurons in dlPFC were coupled to activity in MSTd, while 7% of neurons in MSTd were coupled to activity in dlPFC. Importantly, neither of these regions showed either a 'feedforward' or 'feedback' likelihood of being coupled with area 7a that exceeded 4% (*Figure 4E*).

Four arguments support the fact that unit-to-unit coupling as estimated by the P-GAM reflect a property of neural coding beyond global volume conductance. First, they significantly improved fits of the encoding model. *Figure 4F* shows the cross-validated pseudo-$R^2$ for the encoding model, including all task variables, LFP phases, and unit-to-unit coupling ('coupled model', y-axis), relative to a reduced model lacking the unit-to-unit coupling ('uncoupled model', x-axis). Second, inter-area coupling filters did not show oscillatory behavior. Instead of sinusoidal patterns, we frequently observed gradual increases or decreases in the likelihood of the examined neuron to spike given the activity of another neurons (see *Figure 4—figure supplement 3B* for a 'dictionary' of coupling filters across areas and their relative frequency). Third, while units phase-locked to the LFP in a different region were likely to be phase-locked to LFP in their own area (reflecting the utility of LFPs in coordinating spiking activity across areas), there was no significant change in the likelihood of coupled vs. uncoupled units to be phased-locked to LFPs (*Figure 4—figure supplement 3C*). Lastly, and most importantly, the likelihood of two units being coupled was not random or purely depending on their physical distance, but instead varied as a function of their tuning similarity. *Figure 4—figure supplement 3D* shows that the more similar the tuning functions were between two neurons, the more likely these were to be coupled. The P-GAM was able to reflect this well-established 'like-to-like connectivity' (*Cossell et al., 2015*; *Chettih and Harvey, 2019*) within a naturalistic closed-loop task, even for latent variables, and even in MSTd.

Altogether, both the global encoding profiles and fine-grain unit-to-unit coupling across areas (*Figure 4*) suggested the presence of an MSTd-dlPFC functional subnetwork within this closed-loop virtual navigation task. Given the high probability to encountering neurons in MSTd and dlPFC tuned to the distance and angle from the target, as well as to eye position (*Figure 2*), and given the animals' tendency to keep track of the firefly location with their eyes (*Figure 1*, and *Lakshminarasimhan et al., 2020*), we hypothesized that this MSTd-dlPFC network may reflect the monkey's embodied mnemonic strategy in navigating to hidden targets.

## MST-dlPFC coupling reflect the animals' strategy of tracking hidden targets with their eyes

The monkeys' gaze reflected their belief about the firefly location. Indeed, within a session and across trials, the mean absolute difference between a monkey's eye position and where they ought to be looking if they were perfectly tracking the firefly correlated with steering endpoint error ($r^2$ ± standard deviation across all datasets, 0.36±0.22; for shuffled data, 0.16±0.09, p=5.6 × 10⁻³, paired t-test; see *Lakshminarasimhan et al., 2020*, for the original description of this effect). Further, this relationship also held across sessions, with better target tracking correlating with less bias (slopes closer to 1, see *Figure 1C*), particularly in the angular domain ($r^2$=0.43, p=0.004; radial: $r^2$=0.26, p=0.04; *Figure 5A*), and with an increasing proportion of rewarded trials ($r^2$=0.24, p=0.042). Thus, we may question whether the likelihood of observing unit-to-unit coupling (defined within a session but not within a trial) relates to session-by-session changes in target tracking and/or steering performance.

For well-fit sessions (mean pseudo-$r^2$ >0.05) with at least two units in each of two different areas, we computed the monkey's target tracking index (averaged across the entire trial and then across trials), as well as the probability of units being coupled with others within and across areas, as estimated by the P-GAM. As shown in *Figure 5B* (top right), we observed a strong association between the fraction of units showing MST-to-dlPFC coupling and the monkeys' ability to track the hidden firefly with their eyes ($r^2$=0.63, p=0.003). This association, with more coupling indexing better tracking of the firefly, was also true for the 'feedback' projection from dlPFC-to-MSTd ($r^2$=0.41, p=0.035, *Figure 5B*, bottom left), as well as for dlPFC-to-dlPFC coupling ($r^2$=0.43, p=0.047, *Figure 5B*, bottom right). The other sender-receiver pairings, including MSTd-to-MSTd, did not show a correlation with the ability of the monkey to track the hidden firefly with their eyes (all p>0.08). As shown in *Figure 5—figure supplement 1*, there was no correlation between the unit-to-unit couplings within a session, and either radial

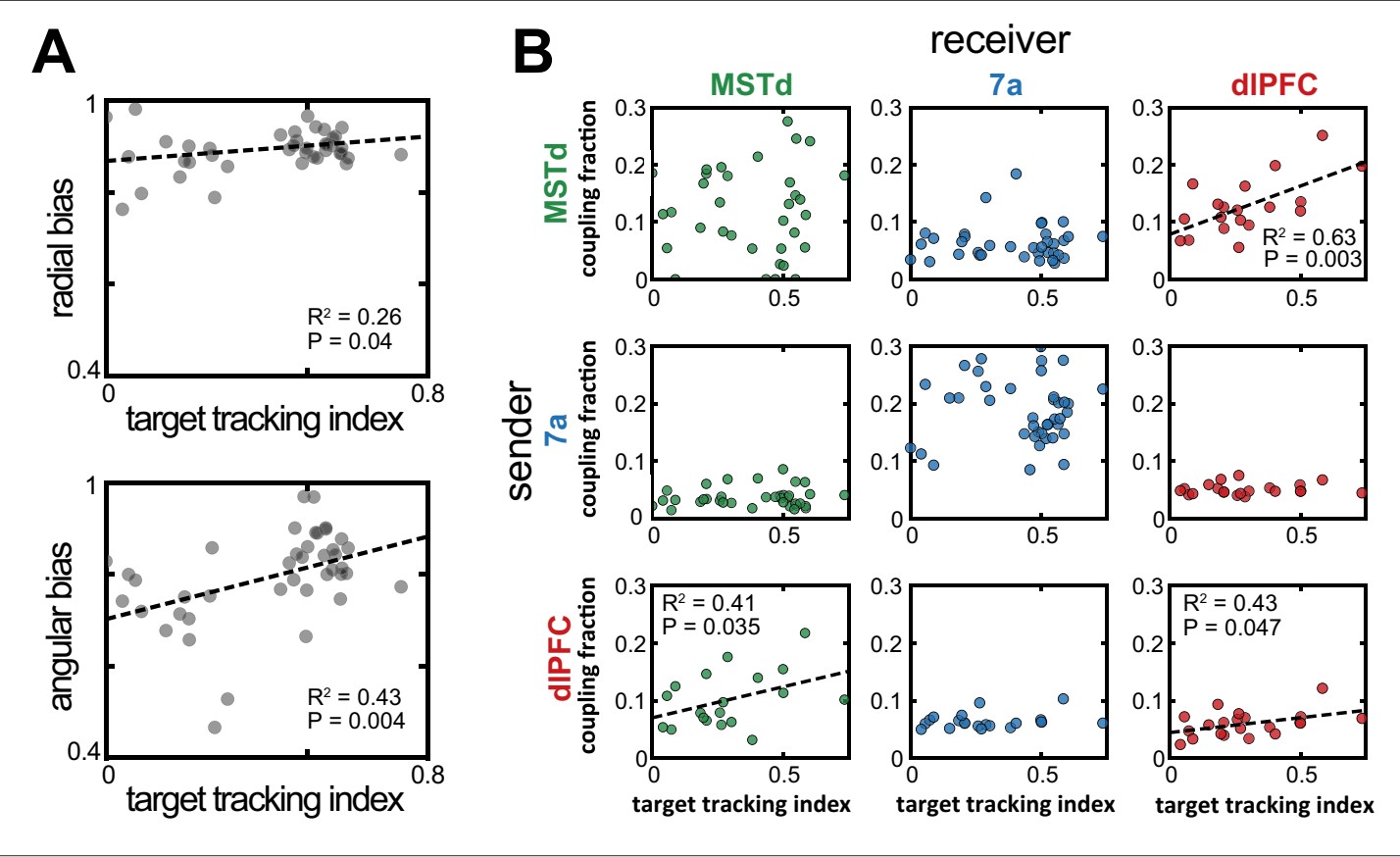

**Figure 5.** Increased dorsomedial superior temporal area (MSTd)-dorsolateral prefrontal cortex (dlPFC) coupling correlated with an increased likelihood of animals keeping track of the invisible fireflies with their eyes. (**A**) Correlation between the target tracking index (x-axis, i.e., tracking the hidden target with their eyes) and the radial (top) or angular (bottom) bias (defined as the slope relating responses and targets, as in *Figure 1*). Only sessions included in the neural analysis in Panel B were included in Panel A. (**B**) Correlation between the target tracking index (x-axis) and the fraction of neurons coupled within the 'sender' region (y-axis). The diagonal shows within area couplings (MSTd-MSTd, 7a-7a, dlPFC-dlPFC), while off-diagonals show across area couplings. $R^2$ and p-values are shown as insets for the significant correlations.

The online version of this article includes the following figure supplement(s) for figure 5:

**Figure supplement 1.** Correlations between fraction of neurons coupled in a 'sender' region and steering behavior.

or angular biases (7a-7a coupling vs. angular bias p=0.06; all other p>0.12). There was similarly no correlation between the proportion of rewarded trials in a session and unit-to-unit coupling probability (all p>0.11, Bonferroni corrected). Overall, therefore, the functional subnetwork between MSTd and dlPFC (*Figure 4*) seemingly reflects the animals' strategy in keeping track of the hidden target with their eyes. In turn, the eye movements (but not MSTd-dlPFC coupling directly) aid in successfully navigating to the location of the hidden target.

## Discussion

The lion's share of our understanding linking spiking activity to goal-directed behavior originates from tasks that bear little resemblance to real-world self-environment interactions. Major differences include the prevalence of closed loops between action and perception in natural behaviors, as well as the necessity to concurrently accomplish multiple sensory and cognitive operations over protracted periods of time. To examine how the brain operates and coordinates among multiple neural nodes under these naturalistic conditions, we recorded simultaneously from three interconnected cortical areas (MSTd, 7a, and dlPFC) while non-human primates detected a target, remembered its location, and path integrated to it by actively sampling from their environment. We focused on the encoding profiles of single units and described four main findings.

First, we found that all task variables were present across all brain areas probed. This multiplexing, however, was not random. The prefrontal cortex showed the greatest proportion of neurons tuned to latent variables. Posterior parietal cortex showed a strong selectivity for sensorimotor variables and the ongoing phase of LFPs. MSTd was prominently tuned to eye position (*Komatsu and Wurtz, 1988*; *Nadler et al., 2009*). Most notoriously, even MSTd, an area traditionally considered a sensory node, showed a considerable number of neurons encoding latent variables, such as the angle or radial distance from origin (i.e., path integration) and to target (i.e., vector coding of spatial goals). In fact, the global encoding profile of MSTd was strikingly akin to that of dlPFC, suggesting this area need not be purely sensory (see *Mendoza-Halliday et al., 2014*, for a similar argument).

Second, we found that MSTd, area 7a, and dlPFC all show evidence for vector-based coding of spatial goals (*Sarel et al., 2017*, also see *Ekstrom et al., 2003*; *Gauthier and Tank, 2018*; *Marigold and Drew, 2017*, for similar evidence in humans, rodents, and cats, respectively). Interestingly, we observed that area 7a showed a majority of neurons coding for locations near the origin or near the target. That is, 7a is likely involved in body-centered state transitions (e.g., from static to moving) and approaching behavior (see *Medendorp and Heed, 2019*, for a similar argument and *Serino, 2019*, for a review on the role of posterior parietal cortex in peri-personal space encoding). dlPFC, on the other hand (and to a lesser extent MSTd), showed a preponderance of units coding for locations near the origin and far from the target. Interestingly, human fMRI work (*Howard et al., 2014*) has suggested that the posterior hippocampus encodes path distance to goals, while the anterior hippo-campus encodes a more abstract Euclidean distance. Together with the current findings, and the fact that anterior and posterior hippocampus respectively project to the prefrontal and parietal cortex (*Strange et al., 2014*) suggests that a circuit including the anterior hippocampus and prefrontal cortex may abstractly plan goal-directed trajectories, while a circuit including the posterior hippocampus and posterior parietal cortex may compute body-centric sensorimotor affordances.

The fact that all areas probed showed a preference for distances near (as opposed to far) from where the monkey started path integrating (i.e., the origin) may suggest a change in the computations undertaken early vs. late within a single trajectory. Namely, early on observers may compute their ongoing distance from the origin, and then switch to computing distance not from origin, but from the target. This switch from first coding displacement from origin to then coding location relative to a target has been previously suggested empirically (*Gothard et al., 1996*) and in recent computational accounts (*Kanitscheider and Fiete, 2017*).

The third major finding relates to the functional organization across MSTd, area 7a, and dlPFC, and again suggests that navigating to a remembered target may involve two interdependent but separable computations: (i) estimating own's own location relative to a starting location (i.e., a land-mark) and (ii) estimating the position of the target and discounting self-location from target location. Indeed, anatomical tracer studies suggest a distributed hierarchy, with recurrent connections between MSTd and 7a, and then recurrent connections between 7a and dlPFC (*Andersen et al., 1990*; *Rozzi et al., 2006*). As expected from anatomy, here we observed that spiking activity in 7a is influenced by LFP fluctuations in MSTd (particularly in the alpha range). In turn, spiking activity in dlPFC and 7a are influenced by LFP fluctuations, respectively, in 7a and dlPFC (particularly in the beta range). In other words, we observed that the coarse channels of communication riding on LFPs are in line with anatomy. Further, the strong tuning of area 7a to sensorimotor variables and the strong tuning of both area 7a and dlPFC (but not MSTd) to the distance and angle from origin suggest that this pathway may be primarily involved in path integration as opposed to the vector coding of spatial goals. In addition to this MSTd-7a-dlPFC pathway, we also observed significant functional coupling between MSTd and dlPFC, independent from 7a. Namely, the likelihood of MSTd neurons being coupled to dlPFC, and of dlPFC neurons being coupled to MSTd, was three and almost two times as large as the unit-to-unit coupling among either of these areas with 7a. These results suggest that despite the lack of direct anatomical connections, MSTd and dlPFC may form a functional subnetwork within this closed-loop navigate-to-target task.

Lastly, we examined the putative functional role of this MSTd-dlPFC subnetwork. As our group has previously reported, monkeys naturally track with their eyes the location of hidden goals during navigation (*Lakshminarasimhan et al., 2020*). Given this task strategy, as well as the prominent tuning of both MSTd and dlPFC to eye position and the distance and angle to the hidden target, we hypoth-esized that the MSTd-dlPFC subnetwork may reflect the monkeys' innate task strategy. Indeed, we

found that the more units were coupled between MSTd and dlPFC (either 'feedforward' or 'feedback'), the better the monkeys tracked the hidden goal with their eyes. Thus, we suggest that the dynamic functional coupling between MSTd and dlPFC may reflect the innate task strategy monkeys adapted in pursuing the goal location with their eyes.

We must acknowledge a number of limitations and areas of ongoing or future experimentation. First, while the task we employed here is more naturalistic than most, further improvements are possible. For instance, virtual and real-world navigation may rely on partially distinct neural codes (*Aghajan et al., 2015*). Thus, it will be interesting to replicate the current experiment while macaques move freely in a 3D environment (e.g., *Mao et al., 2021*). This would also allow for independent eye and head movements (head was restrained here) and thus we could estimate whether eye movements in the current experiment partially reflected intended head movements (as they seemingly do in rodents; *Michaiel et al., 2020*; *Meyer et al., 2020*). Performing a 'firefly task' in a real environment would also suppose a more complex set of visual inputs (e.g., corners, textures, shadows) that could be leveraged in an expanded P-GAM taking visual features as input (see *Parker et al., 2022*, for recent work taking this approach). The second limitation relates to the (necessarily incomplete) sampling of neural areas, and the focus on single units as opposed to population dynamics. We report a functional subnetwork between MSTd and dlPFC based on the similarity of their encoding profiles (though they are of course not identical) and the likelihood of encountering unit-to-unit couplings across these areas. But this functional connection must be subserved by structure (e.g., perhaps a third area we did not record from fluctuating with both MSTd and dlPFC). Thus, in ongoing experiments we have trained rodents to perform the 'firefly task'. This will allow recording from a wider array of neural areas and cortical layers (most of the recordings reported here being from Utah arrays and hence likely from a single layer and of limited independence). Similarly, to further corroborate the functional subnetwork between MST and dlPFC, it will be interesting to examine population dynamics and the possibility that these areas form a functional 'communication subspace' (*Semedo et al., 2019*), adapted to the naturalistic setting of this task (see *Balzani et al., 2022b*). The last limitation relates to causality. The results we report here amount to detecting a correlation (i.e., spikes occurring most often at a given LFP phase and this correlation not being accounted by other sensorimotor, latent, or internal covariates). In future experiments it will be interesting to test for causality within this network, by either demanding observers to fixate elsewhere (*Lakshminarasimhan et al., 2020*), or by directly perturbing this network, for instance by micro-stimulation or optogenetic manipulations.

In conclusion, we demonstrate what may be broad principles of neural computation during closed action-perception loops: (i) mixed yet patterned single cell selectivity, (ii) coding of latent variables even in areas traditionally considered as purely sensory, and (iii) differential coarse (e.g., LFP-spike phase alignment) and fine-grain connectivity between task-relevant areas. Most notoriously, here we indexed the presence of significant noise correlations between MSTd and dlPFC, independently from 7a. We suggest that this coupling between sensory and prefrontal areas may reflect (embodied) task strategies.

## Methods

### Animals and animal preparation

We recorded extracellularly from areas MSTd, area 7a, and dlPFC in three adult male rhesus macaques (*Macaca mulatta*; 9.8–13.1 kg). We collected behavioral and neural data from 27 recording sessions from Monkey Q, 38 sessions from Monkey S, and 17 sessions from Monkey M (see *Figure 2—figure supplement 1* for additional detail regarding the recording locations from each animal). All animals were chronically implanted with a lightweight polyacetal ring for head restraint. Further, for acute recordings, animals were outfitted with a removable grid to guide electrode penetrations. For eye tracking, a subset of monkeys (Monkey Q) were implanted with scleral coils (CNC Engineering, Seattle WA, USA), while eye tracking was performed using a video-based system (ISCAN Inc, Woburn, MA, USA) in the remaining animals (Monkeys S and M). Monkeys were trained using standard operant conditioning procedures to use a joystick to navigate in virtual reality and stop at the location of briefly presented targets, 'fireflies'. All surgeries and procedures were approved by the Institutional Animal Care and Use Committee at Baylor College of Medicine (Protocol A3317-01) and New York University (Protocol number 18-1502) and were in accordance with National Institutes of Health guidelines.

## Experimental setup

Monkeys were head-fixed and secured in a primate chair. A three-chip DLP projector (Christie Digital Mirage 2000, Cypress, CA, USA) rear-projected images onto a 60×60 cm$^2$ screen that was attached to the front of the field coil frame, ~30 cm in front of the monkey. To navigate, the animals used an analog joystick (M20U9T-N82, CTI electronics) with two degrees of freedom to control their linear and angular speeds in a virtual environment. This virtual world comprised a ground plane whose textural elements had limited lifetime (~250 ms) to avoid serving as landmarks. The ground plane was circular with a radius of 70 m (near and far clipping planes at 5 cm and 40 m, respectively), with the subject positioned at its center at the beginning of each trial. Each texture element was an isosceles triangle (base × height: 8.5 × 18.5 cm$^2$) that was randomly repositioned and reoriented at the end of its life-time, making it impossible to use as a landmark. The maximum linear and angular speeds were fixed to 2 m/s and 90 deg/s. The density of the ground plane was 5.0 elements/m$^2$. All stimuli were generated and rendered using C++ Open Graphics Library (OpenGL; Nvidia Quadro FX 3000G accelerator board) by continuously repositioning a virtual camera based on joystick inputs to update the visual scene at 60 Hz. The virtual camera was positioned at a height of 0.1 m above the ground plane. Given the OpenGL approach in re-positioning a virtual camera within a 3D space, depth cues included linear perspective, texture gradient, and motion parallax. Further, the stimulus was rendered as a red-green anaglyph and monkeys wore goggles fitted with Kodak Wratten filters (red #29 and green #61) to view the stimulus. This additionally provided binocular parallax. The binocular crosstalk for the green and red channels was 1.7% and 2.3%, respectively. Spike2 software (Cambridge Electronic Design Ltd., Cambridge, UK) was used to record and store the timeseries of target and animal's location, animal linear and angular velocity, as well as eye positions. All behavioral data were recorded along with the neural event markers at a sampling rate of 833.33 Hz.

## Behavioral task

Monkeys steered to a target location (circular disc of radius 20 cm) that was cued briefly (300 ms) at the beginning of each trial. Each trial was programmed to start after a variable random delay (truncated exponential distribution, range: 0.2–2.0 s; mean: 0.5 s) following the end of the previous trial. The target appeared at a random location between –40 and 40 deg of visual angle, and between 1 and 4 m relative to where the subject was stationed at the beginning of the trial. The joystick was always active, and thus monkeys were free to start moving before the target vanished, or before it appeared. Monkeys typically performed blocks of 750 trials before being given a short break. In a session, monkeys would perform two or three blocks. Feedback in the form of juice reward was given following a variable waiting period after stopping (truncated exponential distribution, range: 0.1–0.6 s; mean: 0.25 s). They received a drop of juice if their stopping position was within 0.6 m from the center of the target. No juice was provided otherwise.

## Neural recording and pre-processing

We recorded extracellularly, either acutely using a 24- or 36-channel linear electrode array (100 μm spacing between electrodes; U-Probe, Plexon Inc, Dallas, TX, USA; MSTd in Monkeys Q and S, and dlPFC in Monkey M) or chronically with multi-electrode arrays (Blackrock Microsystems, Salt Lake City, UT, USA; 96 electrodes in area 7a in Monkey Q, and 48 electrodes in both area 7a and dlPFC in Monkey S). During acute recordings with the linear arrays, the probes were advanced into the cortex through a guide-tube using a hydraulic microdrive. Spike detection thresholds were manually adjusted separately for each channel to facilitate real-time monitoring of action potential waveforms. Recordings began once the waveforms were stable. The broadband signals were amplified and digitized at 20 kHz using a multichannel data acquisition system (Plexon Inc, Dallas, TX, USA) and were stored along with the action potential waveforms for offline analysis. Additionally, for each channel, we also stored low-pass filtered (–3 dB at 250 Hz) LFP signals. For the chronic array recordings, broadband neural signals were amplified and digitized at 30 kHz using a digital headstage (Cereplex E, Blackrock Microsystems, Salt Lake City, UT, USA), processed using the data acquisition system (Cereplex Direct, Blackrock Microsystems) and stored for offline analysis. Additionally, for each channel, we also stored low-pass filtered (–6 dB at 250 Hz) LFP signals sampled at 500 Hz. Finally, copies of event markers were received online from the stimulus acquisition software (Spike2) and saved alongside the neural data.

Spike detection and sorting were performed on the raw (broadband) neural signals using KiloSort 2.0 software (*Pachitariu et al., 2016*) on an NVIDIA Quadro P5000 GPU. The software uses a template matching algorithm both for detection and for clustering of spike waveforms. The spike clusters produced by KiloSort were visualized in Phy2 and manually refined by a human observer, by either accepting or rejecting KiloSort's label for each unit. In addition, we computed three isolation quality metrics: inter-spike interval violations (ISIv), waveform contamination rate (cR), and presence rate (PR). ISIv is the fraction of spikes that occur within 1 ms of the previous spike. cR is the proportion of spikes inside a high-dimensional cluster boundary (by waveform) that are not from the cluster (false positive rate) when setting the cluster boundary at a Mahalanobis distance such that there are equal false positives and false negatives. PR is 1 minus the fraction of 1 min bins in which there is no spike. We set the following thresholds in qualifying a unit as a single unit: ISIv <20%, cR <0.02, and PR >90%.

## Analyses

### Behavior

The location of targets and monkey's endpoints were expressed in polar coordinates, with a radial distance (target = $r$, response = $\widetilde{r}$) and eccentricity from straight ahead (target = $\theta$; response = $\widetilde{\theta}$). On a subset of trials (~5%) animals stopped within 0.5 m of the origin (range of targets, 1–4 m). Similarly, on a subset of trials (~13%) animals did not stop during the course of a trial (max duration = 7 s). These trials were discarded before further behavioral analyses. As we have observed before (*Lakshminarasimhan et al., 2018*; *Lakshminarasimhan et al., 2020*; *Noel et al., 2020*; *Noel et al., 2021*), a linear model with multiplicative gain accounted well for the observed data (average $R^2$=0.72). Thus, we used the slopes of the corresponding linear regressions as a measure of bias. Note that in this scheme a slope of one indicates no bias (i.e., targets and endpoints lie along the identity line), whereas slopes smaller than one indicate a bias wherein animals undershoot targets (either in radial or angular distance).

### Poisson generalized additive model

The P-GAM (https://github.com/savin-lab; *Balzani, 2020a*) defines a non-linear mapping between spike counts of a unit $y_t \in N_0$ and a set of continuous covariates $x_t$ (angular and linear velocity and acceleration, angular and linear distance travelled, angular and linear distance to target, and LFP instantaneous phase across different frequency ranges), and discrete events $z_t$ (time of movement onset/offset, target onset, reward delivery, and the spike counts from simultaneously recorded units). As such, inputs to the P-GAM were of three types. First, the spike counts of the unit to be modeled, at a 6 ms resolution (i.e., the number of spikes within 6 ms windows, no baseline correction). Second, the continuous, discrete, and neural covariates, which were also sampled at a 6 ms resolution. The last input type were a set of 15 'knots' per covariate, defining the nodes of eventual tuning functions. The location of knots were defined as to (i) cover the range of a given input variable from the 2nd to the 98th percentile with (ii) equi-probable knots (each knot covering the same probability mass). See (https://github.com/BalzaniEdoardo/PGAM/blob/master/PGAM%20Tutorial.ipynb; *Balzani, 2022c*) for a comprehensive tutorial.

The unit log-firing rate is modeled as a linear combination of arbitrary non-linear functions of the covariates,

$$log\mu = \sum_j f_j\left(x_j\right) + \sum_k f_k * z_k \tag{1}$$

where * is the convolution operator, and the unit spike counts are generated as Poisson random variables with rate specified by *Equation 1*.

Input specific non-linearities $f\left(\cdot\right)$ are expressed in terms of flexible B-splines, $f\left(\cdot\right) \approx \beta \cdot b\left(\cdot\right)$ and are associated with a smoothness enforcing penalization term controlled by a scale parameter $\lambda_f$,

$$PEN\left(f, \lambda_f\right) = \frac{-1}{2}\lambda_f \beta^T S_f \beta, S_f = \int b^{''} \cdot b^{''T} dx \tag{2}$$

The larger $\lambda_f$, the smoother the model. This penalization terms can be interpreted as Gaussian priors over model parameters. The resulting log-likelihood of the model takes the form,

$$L\left(y\right) = log\, p\left(y|x, z, \beta\right) + \sum_f PEN\left(f, \lambda_f\right) \qquad (3)$$

with $y \in R^T$ being the spike counts of the unit, $x \in R^{J \times T}$ being the continuous task variables, $z \in R^{K \times T}$ being the discrete task events, $T$ being the time points, $\beta$ being the collection of all B-spline coefficients, and $p\left(\cdot\right)$ the Poisson likelihood. Both parameters $\beta$ and the hyperparameters $\lambda$ are learned from the data by an iterative optimization procedure that switches between maximizing *Equation 3* as a function of the parameters and minimizing a cross-validation score as a function of the hyperparameters (see *Balzani et al., 2020b*, for further details).

The probabilistic interpretation of the penalization terms allowed us to compute a posterior distribution for the model parameters, derive confidence intervals with desirable frequentist coverage properties, and implement a statistical test for input variable inclusion that selects a minimal subset of variables explaining most of the variance. In *Figure 2—figure supplement 2*, we demonstrate that this approach will appropriately select variables contributing to the spike trials of individual neurons. Further, we show that including all variables in the model (hundreds to thousands of parameters, given the cell-to-cell coupling) does not outperform the selected (i.e., 'reduced') model, which typically has an order of magnitude less parameters. The approach circumvents traditional model-comparison-based variable selection, which would be unfeasible in a fully coupled model with hundreds of covariates.

To show the stability in the estimated tuning functions, *Figure 2—figure supplement 10* shows the fraction of units tuned to a given task variable as a function of brain area, and as a function of whether odd or even trials were fit to the P-GAM. Namely, we fit half of the dataset each time and show that the fraction of neurons tuned to a given task variable was the same regardless of whether we fit the odd numbered trials or the even numbered trials. Similarly, we index the 'preferred' distances and angles from origin and to target (*Figure 3*) as defined by the peak of tuning functions. In *Figure 3B* we sort neurons according to their preferred distances or angles in one subset of trials (i.e., 'even' trials) and plot the normalized responses in the other subset of trials ('odd' trials). *Figure 3B* therefore demonstrates that not only the fraction of neurons tuned to different variables was stable, but the estimated tuning functions were as well.

## Pseudo-$R^2$

Fit quality was assessed via the pseudo-$R^2$ on subset of held-out test trials (20% of the total trials, not used for inferring model parameters). Pseudo-$R^2$ is a goodness of fit measure that is suitable for models with Poisson observation noise (*Colin Cameron and Windmeijer, 1997*). The score is computed as:

$$pseudo\,R^2 = 1 - \frac{L\left(y\right) - L\left(\hat{y}\right)}{L\left(y\right) - L\left(\bar{y}\right)} \qquad (4)$$

with $L\left(y\right)$ being the likelihood of the true spike counts, $L\left(\hat{y}\right)$ being the likelihood of the P-GAM model prediction, and $L\left(\bar{y}\right)$ being the likelihood of a Poisson null-model (constant mean rate). It can be interpreted as the fraction of the maximum possible likelihood (normalized by the null model) that the fit achieves. The score is 0 when the GAM fits are no more likely than the null model, 1 when it perfectly matches the data, and rarely can be negative (for test-set data, 3% of the recorded units) when overfitting occurs. In this latter case we excluded the unit from analysis. Empirically, the pseudo-$R^2$ is a stringent metric and ranges in values that are substantially lower than the standard $R^2$ when both are applicable (*Domencich and McFadden, 1975*) and depends on the firing statistics of the area recorded (such as mean firing rates and variance), the type of covariates, and their number (*Benjamin et al., 2018*). Our average score of 0.072 is about three times better than standard GLM performance even in areas such as primary somatosensory and motor areas (*Benjamin et al., 2018*).

## Speed and direction discrimination index

To allow for comparison with prior reports studying optic flow processing within the cadre of two-alternative forced-choice tasks, we compute the discrimination index for speed (i.e., linear velocity) and direction (i.e., angular velocity) in MSTd, 7a, and dlPFC. The discrimination index (DDI) was defined as:

$$DDI = \frac{Rmax - Rmin}{Rmax - Rmin + 2\sqrt{\frac{SSE}{(N-M)}}} \tag{5}$$

where $R_{max}$ and $R_{min}$ are the maximum and minimum response from a tuning function, $SSE$ is the sum of squared errors around the mean responses, $M$ is the number of stimulus directions, and $N$ is the total number of observations. In the context of the current naturalistic experiment, we may bin linear and angular velocities and compute tuning functions defining $R_{max}$ and $R_{min}$. The number of bins (here, we used 15 nodes, as defined by the P-GAM) defines $M$. To estimate $SSE$ and $N$ we must define trials wherein the full gamut of linear and angular velocities are experienced. To facilitate direct comparison with *Chen et al., 2008*, we divided our recordings in 80 segments, the mean number of trials ($N$) in *Chen et al., 2008*. Lastly, we computed DDI according to *Equation 5*.

## Clustering

For *Figure 4A and B*, clustering was performed by employing a spectral clustering algorithm based on the Jaccard distance metric (*Shi and Malik, 2000*). This approach finds the $k$ eigenvectors to split Jaccard distances $k$ ways. Different values of $k$ (≥3) resulted in conceptually identical findings. The results reported in the main text were generated for the full dataset, and thus a 4254 (231 MSTd neuron + 3200 7a neurons + 823 dlPFC neurons) × 17 (task variables) input matrix (where each entry denotes whether a particular neuron was significant or not for a particular task variable). In *Figure 4— figure supplement 1A* we confirm that the results reported in the main text were not driven by the fact that we had a considerably larger number of 7a neurons than other. Namely, we perform 10,000 iterations of spectral clustering while subsampling from 7a and dlPFC (without replacement) to match the number of units in MSTd (231). At each iteration, we compute the ratio of the size of the largest cluster, to the total number of units for a given area (i.e., 231). Results confirm that MSTd and dlPFC neurons were most often assigned to a single cluster, while neurons in 7a were most readily assigned to one of three clusters.

For *Figure 4C* all tuning functions of a neuron were stacked together, so each neuron was represented by a vector of dimension $\sum n_i$, where $n_i$ is the dimension of the $i$ th tuning function. Then, the high-dimensional matrix (neurons × $\sum n_i$) was reduced to a matrix (neurons × $m$) by PCA projection, where $m$ is the number of PCs that explains >90% of the variance. Finally, the matrix was further projected onto a 2D manifold using UMAP (*McInnes et al., 2020*) and clustered using DBSCAN (*Ester, 1996*). This clustering method automatically determined the number of clusters on the basis of the spatial density of the UMAPs. Clusters depicted in *Figure 3D*, *Figure 3—figure supplements 1–4*, and *Figure 4—figure supplement 3B* were equally determined by DBSCAN. As for *Figure 4A and B*, for *Figure 4C* we confirmed that the results reported in the main text were not driven by the unequal number of neurons recorded from in MSTd, 7a, and dlPFC. Namely, we perform 500 iterations of the procedure described above (i.e., stacking of tuning functions, then PCA, then UMAP) while subsampling from 7a and dlPFC to match the number of units present in MSTd. At each iteration, we compute the distance between the MSTd and dlPFC centroid in UMAP space, and between the MSTd and 7a centroid. We then compute the ratio of these distances, which are depicted in *Figure 4—figure supplement 1B*.

For *Figure 4—figure supplement 2*, mutual information was computed as the difference of the entropy of spike counts $H(Y)$, and the entropy of the counts conditioned on the stimulus $H(Y|S)$,

$$I(Y, S) = H(Y) - H(Y|S) \tag{6}$$

To estimate these quantities, we assumed that the counts $Y$ are Poisson distributed with rate parameter $\lambda$ equal to the mean firing rate of the unit. The stimulus was discretized and its distribution approximated as a binomial, while the conditional distribution of the counts given the stimuli $p(Y|S)$ is given by the P-GAM generative model. Finally, we computed $Y$ as:

$$H(Y) = -\sum_y p(y)\, logp(y)\,, \text{and } H(Y|S) = -\sum_s p(s) \sum_y p(y|s)\, logp(y|s) \tag{7}$$

## Coupling filters

Coupling filters (and the corresponding inferential statistics) were determined via the P-GAM. Within area coupling filters were set to a duration of 36 ms, and across area filters were set to a duration of 600 ms. For the within-area coupling probability reported in *Figure 4E*, we corrected for the effects

of unit distance (i.e., *Figure 4—figure supplement 3A*) by first fitting a brain region-specific logistic regression. Specifically, we expressed coupling probability as a non-linear function of electrode distance as follows:

$$p\left(c=1\right) = logit^{-1}\left(f\left(d\right)\right) \tag{8}$$

with $c$ being a binary variable taking value 1 for significant coupling and 0 otherwise, $d$ being the electrode distance, and $f$ being a non-linear function expressed in terms of B-splines. Each brain area was fit independently, and the coupling probability in *Figure 4E* was set as the model prediction for a distance of 500 μm.

## Data and code availability

Data and code are available at: https://osf.io/d7wtz/. Code and tutorials for utilizing the P-GAM are additionally available at: https://github.com/BalzaniEdoardo/PGAM, (copy archived at swh:1:rev:deaaef-66ccff5e667fcfbbc11c3de75dafea5be4; *Balzani, 2022a*).

## Acknowledgements

The authors thank Jing Lin and Jian Chen for programming the experimental stimulus. We also thank Roozbeh Kiani for his surgical expertise during the Utah array implantations. The work was funded by a 1U19 NS118246, 1R01 NS120407, and 1R01 DC004260 from NIH to DEA, as well as by 1R01MH125571 from NIH, the National Science Foundation under NSF Award No. 1922658, and a Google faculty award to CS.

## Additional information

### Funding

| Funder | Grant reference number | Author |
|---|---|---|
| National Institutes of Health | 1U19 NS118246 | Dora E Angelaki |
| National Institutes of Health | 1R01 NS120407 | Dora E Angelaki |
| National Institutes of Health | 1R01 DC004260 | Dora E Angelaki |
| National Institutes of Health | 1R01MH125571 | Cristina Savin |
| National Science Foundation | 1922658 | Cristina Savin |
| Google faculty award | | Cristina Savin |

The funders had no role in study design, data collection and interpretation, or the decision to submit the work for publication.

### Author contributions

Jean-Paul Noel, Conceptualization, Data curation, Software, Formal analysis, Validation, Investigation, Visualization, Methodology, Writing - original draft, Writing – review and editing; Edoardo Balzani, Conceptualization, Resources, Data curation, Software, Formal analysis, Validation, Investigation, Visualization, Methodology, Writing – review and editing; Eric Avila, Conceptualization, Data curation, Investigation, Methodology; Kaushik J Lakshminarasimhan, Conceptualization, Data curation; Stefania Bruni, Panos Alefantis, Methodology; Cristina Savin, Supervision, Validation, Investigation, Project administration, Writing – review and editing; Dora E Angelaki, Conceptualization, Supervision, Funding acquisition, Project administration, Writing – review and editing

### Author ORCIDs

Jean-Paul Noel (ID) http://orcid.org/0000-0001-5297-3363

Edoardo Balzani [ID] http://orcid.org/0000-0002-3702-5856
Dora E Angelaki [ID] http://orcid.org/0000-0002-9650-8962

### Ethics

All surgeries and procedures were approved by the Institutional Animal Care and Use Committee at Baylor College of Medicine and New York University and were in accordance with National Institutes of Health guidelines.

### Decision letter and Author response

Decision letter https://doi.org/10.7554/eLife.80280.sa1
Author response https://doi.org/10.7554/eLife.80280.sa2

---

## Additional files

### Supplementary files

• MDAR checklist

### Data availability

Data and code are available at: https://osf.io/d7wtz/.

The following dataset was generated:

| Author(s) | Year | Dataset title | Dataset URL | Database and Identifier |
|---|---|---|---|---|
| Noel J, Balzani E | 2022 | latent variables in sensory, parietal, and frontal cortices | https://osf.io/d7wtz | Open Science Framework, d7wtz |

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
