## [Editor Report]

This important study investigates distributed neural coding across the three brain areas MST, 7a, and dlPFC in monkeys carrying out a novel behavioural paradigm with a naturalistic closed action-perception-loop developed by the same group previously. The convincing model-based analysis discerns potential influences (e.g. task variables, hidden variables) on firing rates and supports the claim of task-specific sub-networks being formed. The authors provide an important first step to unravel potential drivers of dynamic activity in distributed networks during recurrent action-perception-loops, which should be augmented by future analyses of, for instance, the contribution of changing visual input, especially as the recordings stem from areas involved in processing optical flow, and of signals across different circuit elements like cortical layers.

---

## [Decision Letter]

**Decision letter after peer review:**

Thank you for submitting your article "Coding of latent variables in sensory, parietal, and frontal cortices during virtual closed-loop navigation" for consideration by *eLife*. Your article has been reviewed by 3 peer reviewers, one of whom is a member of our Board of Reviewing Editors, and the evaluation has been overseen by Joshua Gold as the Senior Editor. The following individuals involved in review of your submission have agreed to reveal their identity: Marieke Schölvinck (Reviewer #2) and Sujaya Neupane (#3).

The reviewers have discussed their reviews with one another, and the Reviewing Editor has drafted this letter to help you prepare a revised submission.

Essential revisions:

1) Heat maps of preferred angle and distance in Figure 3 must be cross-validated across trials to show data reliability. Maybe the authors have done that and these are cross-validated plots. If so, it is not mentioned in the methods anywhere. Please verify the ordering of preferred angle and distance hold up with cross-validation. One can get a spurious, evenly tiled coding of any continuous variable if one takes a random matrix (say mean FR x variable value), normalizes each row (FR) and sorts the column by peak location (i.e. preferred variable value) for each row.

2) We are a bit confused by the distribution of preferred latent variables (Figure 3). For e.g. travelled distance and distance to target are anti-correlated. Isn't it trivial that the preferred coding would appear bimodal across a neural population if some neurons are coding for one and some for the other? For a travelled distance coding neuron, there is nothing to code at the onset since distance travelled is 0 and vice versa for distance to the target coding neuron at the offset.

Related to #2, it would be helpful to see PSTH examples of single neurons that code for travelled distance and those that code for distance to target. PSTH would be obtained by averaging across trials, binned over a range of trial-lengths (e.g. bin1: short trial length, bin2: medium trial length, bin3: long trial length). We would expect clear differences in firing rate at the onset for distance to target coders and at the offset for travelled distance coders. It is difficult to see this in the presented rasters, although according to P-GAM results, that should be the case.

3) It would be helpful to provide a few examples of LFP traces and their filtered form along with spike times to appreciate the phase modulations apparent in their statistical modelling results (Figure 2F).

4) Tuning strength.

From the manuscript it is difficult to judge how representative the different neuronal populations for each area are and to what extent their selectivity differs. The analysis and it variables are quite complex to follow. It would be helpful for the reader to understand how some of them relate to more traditional measures. It is great that the focus on single neurons allows this comparison.

What does "tuned" in this context mean in terms of strength and selectivity?

How many neurons would pass a minimal response criterion like 10 spikes/s. Would these show stronger tuning or correlations?

Figure 2E: "E. Responses from an example MSTd, 7a, and dlPFC neuron (black), aligned to temporal task variables (e.g., time of movement onset and offset), or binned according to their value in a continuous task variable (e.g., linear velocity)." It would be helpful to give the x-axis for each line for the reader to be able to ascertain what the nature of the scale and the range of variables are over which the firing rate changes are depicted.

Could the authors derive from their data one of the traditional measures used for MST, like a direction tuning index? A direct comparison with previous studies could help understand the nature of the sampled pool, particularly (but not exclusively) for areas when smaller neuronal samples, like MST.

5) Eye movements and visual input.

Another issue is the extent that it seems difficult to distinguish the effect of eye position from that of the background stimulus flow patterns, which of course must differ in direction and element size when animals fixate at different locations on the screen. To what extent was this visual input to neurons correlated with "latent variables" like latent distance and angle to target (latent spatial goal)?

In order to dissociate the contribution of eye position and task from visual input, do the authors have data on a passive viewing control condition, in which the animal fixates and the visual pattern is played back to animals exactly as if in an active one? How do neural responses compare across the three areas?

Could the authors discuss in the paper how the visual input is (or not) included in the model?

6) MSTd and dlPFC coupling.

a) As the animals were head-fixed, eye position would compensate in some cases the animals might have moved its head position (for instance to keep track of the target). Both, MSTd and dlPFC encoded eye position. Could the close coupling of MSTd and dlPFC be linked to this element of the task?

b) The authors claim that areas MSTd and dlPFC form a functional sub-network together, on the basis of similarity in the fractions of neurons tuned to certain variables, and the distribution of the preferred value of some of these variables. However, the fractions of neurons tuned to the latent variables in MSTd and dlPFC (see Figure 2F) are actually quite different. Perhaps the authors could comment on this.

c) When there was stronger MSTd-to-dlPFC coupling and better tracking of the hidden firefly with the eyes (Figure 5B), was the performance of the monkey also better (i.e. more hits)?

7) Sampling of areas.

a) Area 7a was exclusively sampled with chronic rather than moveable probes. It has also the largest number of "single units".

To what extent are these single units independent?

Could a sampling bias in these probes (part of 7a; layers) affect the results, especially when it comes to coupling. Please include in the discussion.

b) The number of recorded neurons in the three areas differs greatly: 231 units in MSTd, 823 units in dlPFC, and 3200 units in area 7a. Yet many conclusions in the paper rely on neuronal numbers: the fractions of neurons tuned to certain sensorimotor and latent variables differ between the areas, the variables explaining the firing rates cluster differently in the neurons of the three areas, and both the coarse LFP connectivity and the fine unit-to-unit coupling within areas differ. Especially the clustering results might depend on the number of recorded neurons: the fact that almost all MSTd and dlPFC neurons are categorized as belonging to the same cluster, whereas the area 7a neurons appear in three distinct clusters, could be caused by the much larger number of recorded neurons in area 7a. Also unit-to-unit coupling is more likely to show up in the data with a much larger number of recorded neurons. The data could be corrected for these differences in number of recorded neurons.

8) Lateralisation.

To what extent played the lateralization of the recording and task a role for neuronal response? This applies relative to brain hemisphere, body and eye position? Where in each monkey did the recordings take place? Which hand(s) did each monkeys use for the choice stick?

How was the lateralisation included in the model?

Please comment with regards to responses in MST, 7A, and dPFC and add information to the manuscript.

Specifically, it is unclear from Suppl Figure 1 whether within a particular monkey, some recording sites were interhemispheric, or whether within one monkey, all recordings were done in the same hemisphere. This of course has significant consequences for the effects of ongoing LFP and unit-to-unit coupling.

9) Data fed into the P-GAM model.

a) The P-GAM model is a great analysis tool for these kinds of data. However, the variables that the authors put into it are conceptually very different from each other. There are purely external task variables such as target onset and offset, latent variables such as distance to target that require knowledge of one's own position in space, and purely internal brain dynamics variables such as coupling to the LFP in another area. In that light, the finding of 'many variables contributing to the responses' is not surprising; all neurons in the brain are probably influenced both by external variables and internal brain dynamics. Maybe the authors could comment on the different nature of their variables and how that impacts their results.

b) Given that the sensorimotor and latent variables going into the G-PAM model are so crucial for the story, could you make a figure where you visualize them? This could maybe be added to Figure 1A. Also, 'radial bias' and 'angular bias' (in Fig1D) could be visualized here.

c) Quantification of electrophysiological activity processing that is fed into the P-GAM model is not entirely clear.

More details about the preprocessing of these data are required, for example, are the SUA baselined using pre-stimulus presentation activity? Are the LFP baselined as well? And how similar are the pooled responses within each area and across? This would allow the reader to spot possible problems when computing further neuronal properties, that could bias the main paper result:

An example is the tuning of the neurons to the phase of ongoing oscillation (Β, Α, Theta). There are a number of papers attempting to optimize methods to measure spike field coherency, e.g. the PPC pairwise phase consistency (Vinck et al., 2010). This method gives an estimation independent of spike count and LFP amplitudes (both parameter vary of course widely across time, tasks, subjects, areas…).

Here, it seems these two parameters are not considered and could lead to artefacts in the coupling results presented. The authors use temporal correlations to approximate coupling between spike/spike, and spike/LFP-phase. Correlation methods can potentially lead to artefacts and overestimations of coupling strength.

In their methods, the author state to 'bin spiking activity across 8ms window' prior to feeding this activity to the P-GAM. It means that 1 spike corresponds to an averaged 8ms time window. If you now try to calculate the dependency of this single spike to a specific phase of a β (30Hz) , the α (12Hz) and theta (4Hz) oscillation, it means that the chance level of assigning the binned spike to a particular phase differs considerably. Therefore, the statistical power of this analysis would decrease for higher frequency. It seems that the authors do not apply any correction.

*Reviewer #1 (Recommendations for the authors):*

1) From the manuscript it is difficult to judge how representative the different neuronal populations for each area are and to what extent their selectivity differs. The analysis and its variables are quite complex to follow. It would be helpful for the reader to understand how some of them relate to more traditional measures. It is great that the focus on single neurons allows this comparison.

What does "tuned" in this context mean in terms of strength and selectivity?

How many neurons would pass a minimal response criterion like 10 spikes/s. Would these show stronger tuning or correlations?

Figure 2E: "E. Responses from an example MSTd, 7a, and dlPFC neuron (black), aligned to temporal task variables (e.g., time of movement onset and offset), or binned according to their value in a continuous task variable (e.g., linear velocity)." It would be helpful to give the x-axis for each line for the reader to be able to ascertain what the nature of the scale and the range of variables are over which the firing rate changes are depicted.

Could the authors derive from their data one of the traditional measures used for MST, like a direction tuning index? A direct comparison with previous studies could help understand the nature of the sampled pool, particularly (but not exclusively) for areas when smaller neuronal samples, like MST.

2) Another issue is the extent that it seems difficult to distinguish the effect of eye position from that of the background stimulus flow patterns, which of course must differ in direction and element size when animals fixate at different locations on the screen. To what extent was this visual input to neurons correlated with "latent variables" like latent distance and angle to target (latent spatial goal)?

In order to dissociate the contribution of eye position and task from visual input, do the authors have data on a passive viewing control condition, in which the animal fixates and the visual pattern is played back to animals exactly as if in an active one? How do neural responses compare across the three areas?

Could the authors discuss in the paper how the visual input is (or not) included in the model?

3) As the animals were head-fixed, eye position would compensate in some cases the animals might have moved its head position (for instance to keep track of the target). Both, MSTd and dlPFC encoded eye position. Could the close coupling of MSTd and dlPFC be linked to this element of the task?

4) Area 7a was exclusively sampled with chronic rather than moveable probes. It has also the largest number of "single units".

To what extent are these single units independent?

Could a sampling bias in these probes (part of 7a; layers) affect the results, especially when it comes to coupling. Please include in the discussion.

5) To what extent played the lateralization of the recording and task a role for neuronal response? This applies relative to brain hemisphere, body and eye position? Where in each monkey did the recordings take place? Which hand(s) did each monkeys use for the choice stick?

How was the lateralisation included in the model?

Please comment with regards to responses in MST, 7A, and dPFC.

6) Quantification of task parameters are quite clear, this is not entirely the case for electrophysiological activity processing that they feed into their P-GAM model.

More details about the preprocessing of these data are required, for example, are the SUA baselined using pre-stimulus presentation activity? Are the LFP baselined as well? And how similar are the pooled responses within each area and across? This would allow the reader to spot possible problems when computing further neuronal properties, that could bias the main paper result:

An example is the tuning of the neurons to the phase of ongoing oscillation (Β, Α, Theta). There are a number of papers attempting to optimize methods to measure spike field coherency, e.g. the PPC pairwise phase consistency (Vinck et al., 2010). This method gives an estimation independent of spike count and LFP amplitudes (both parameter vary of course widely across time, tasks, subjects, areas…).

Here, it seems these two parameters are not considered and could lead to artefacts in the coupling results presented. The authors use temporal correlations to approximate coupling between spike/spike, and spike/LFP-phase. Correlation methods can potentially lead to artefacts and overestimations of coupling strength.

In their methods, the author state to 'bin spiking activity across 8ms window' prior to feeding this activity to the P-GAM. It means that 1 spike correspond to an averaged 8ms time window. If you now try to calculate the dependency of this single spike to a specific phase of a β (30Hz) , the α (12Hz) and theta (4Hz) oscillation, it means that the chance level of assigning the binned spike to a particular phase differs considerably. Therefore, the statistical power of this analysis would decrease for higher frequency. It seems that the authors do not apply any correction.

*Reviewer #2 (Recommendations for the authors):*

– I am missing a clear motivation for recording in the three areas that you chose. Could you maybe elaborate on this a bit more in the introduction?

– Given that the sensorimotor and latent variables going into the G-PAM model are so crucial for the story, could you make a figure where you visualize them? This could maybe be added to Figure 1A. Also, 'radial bias' and 'angular bias' (in Fig1D) could be visualized here.

– In Figure 1C, you have added 'slope=bias', whereas technically, it is 'deviation from slope=bias'.

– The legend of Figure 2 is extremely long and contains a lot of information that does not pertain directly to the figure. I suggest that the part '(The direct comparison of the goodness-of-fit….the complexity of their areas and tasks, reaches)' in Fig2D is taken out and added to the text somewhere else.

– It is unclear from Suppl Figure 1 whether within a particular monkey, some recording sites were interhemispheric, or whether within one monkey, all recordings were done in the same hemisphere. This of course has significant consequences for the effects of ongoing LFP and unit-to-unit coupling.

– In Fig2F, you show fractions of neurons tuned to the several variables of the G-PAM model, and in Fig4D, you show proportions of neurons phase-locked to LFP phases in other areas. I might have missed it, but I didn't see any quantification of how strong the tuning was, and how strong the phase-locking.

– When there was stronger MSTd-to-dlPFC coupling and better tracking of the hidden firefly with the eyes (Figure 5B), was the performance of the monkey also better (i.e. more hits)?

– There are a few spelling mistakes throughout the paper (psueudo-R on p.6; tunning on p.7)

*Reviewer #3 (Recommendations for the authors):*

1. Heat maps of preferred angle and distance in Figure 3 must be cross-validated across trials to show data reliability. Maybe the authors have done that and these are cross-validated plots. If so, it is not mentioned in the methods anywhere. Please verify the ordering of preferred angle and distance hold up with cross-validation. One can get a spurious, evenly tiled coding of any continuous variable if one takes a random matrix (say mean FR x variable value), normalizes each row (FR) and sorts the column by peak location (i.e. preferred variable value) for each row.

2. I am a bit confused by the distribution of preferred latent variables (Figure 3). For e.g. travelled distance and distance to target are anti-correlated. Isn't it trivial that the preferred coding would appear bimodal across a neural population if some neurons are coding for one and some for the other? For a travelled distance coding neuron, there is nothing to code at the onset since distance travelled is 0 and vice versa for distance to the target coding neuron at the offset.

3. Related to #2 above, it would be helpful to see PSTH examples of single neurons that code for travelled distance and those that code for distance to target. PSTH would be obtained by averaging across trials, binned over a range of trial-lengths (e.g. bin1: short trial length, bin2: medium trial length, bin3: long trial length). I would expect clear differences in firing rate at the onset for distance to target coders and at the offset for travelled distance coders. It is difficult to see this in the presented rasters, although according to P-GAM results, that should be the case.

4. It would be helpful to provide a few examples of LFP traces and their filtered form along with spike times to appreciate the phase modulations apparent in their statistical modelling results (Figure 2F).

---

## [Author Response]

Essential revisions:1) Heat maps of preferred angle and distance in Figure 3 must be cross-validated across trials to show data reliability. Maybe the authors have done that and these are cross-validated plots. If so, it is not mentioned in the methods anywhere. Please verify the ordering of preferred angle and distance hold up with cross-validation. One can get a spurious, evenly tiled coding of any continuous variable if one takes a random matrix (say mean FR x variable value), normalizes each row (FR) and sorts the column by peak location (i.e. preferred variable value) for each row.

We thank the reviewer for this suggestion. We had previously not cross-validated those plots, but have now. We fit the P-GAM to half the dataset (either “odd” or “even” trials). Then, for the figure referenced by the reviewer, we find the distance and angle that most strongly drives a particular neuron (given that the neuron is significantly tuned to the variable), as indicated by their tuning function (i.e., peak of the tuning function). Neurons were then sorted based on their preferred angle/distance in “even” trials, and the tuning functions generated by the “odd” trials are plotted as a heatmap. As shown in Figure 3, the results originally reported hold after cross-validation.

We have amended the figure caption and the methods section to include this information.

Figure caption:

“Heatmaps showing neural responses (y-axis) sorted by preferred angles from origin (top), angle to target (2^nd^ row), distance from origin (3^rd^ row), and distance to target (bottom row) for MSTd (green), 7a (blue) and dlPFC (red) in monkey S (data simultaneously recorded). Darker color indicates higher normalized firing rate. Neurons were sorted based on their preferred distances/angles in even trials and their responses during odd trials is shown (i.e., sorting is cross-validated, see Methods).”

Methods:

“To show the stability in the estimated tuning functions, Figure 2 – supplement 10 shows the fraction of units tuned to a given task-variable as a function of brain area, and as a function of whether odd or even trials were fit to the P-GAM. Namely, we fit half of the dataset each time and show that the fraction of neurons tuned to a given task variable was the same regardless of whether we fit the odd numbered trials, or the even numbered trials. Similarly, we index the “preferred” distances and angles from origin and to target (Figure 3) as defined by the peak of tuning functions. In Figure 3B we sort neurons according to their preferred distances or angles in one subset of trials (i.e., “even” trials) and plot the normalized responses in the other subset of trials (“odd” trials). Figure 3B, therefore demonstrates that not only the fraction of neurons tuned to different variables was stable, but the estimated tuning functions were as well.”

2) We are a bit confused by the distribution of preferred latent variables (Figure 3). For e.g. travelled distance and distance to target are anti-correlated. Isn't it trivial that the preferred coding would appear bimodal across a neural population if some neurons are coding for one and some for the other? For a travelled distance coding neuron, there is nothing to code at the onset since distance travelled is 0 and vice versa for distance to the target coding neuron at the offset.Related to #2, it would be helpful to see PSTH examples of single neurons that code for travelled distance and those that code for distance to target. PSTH would be obtained by averaging across trials, binned over a range of trial-lengths (e.g. bin1: short trial length, bin2: medium trial length, bin3: long trial length). We would expect clear differences in firing rate at the onset for distance to target coders and at the offset for travelled distance coders. It is difficult to see this in the presented rasters, although according to P-GAM results, that should be the case.

We apologize for the confusion and recognize this deserved a more detailed explanation. The distance from origin and to target are not correlated (in fact, if they were fully correlated the P-GAM would be under-specified). Within a trial, these distances could be correlated (e.g., if the target were straight ahead and the animal traveled in a perfectly straight line), but need not. For instance, if the target were at 200 cm straight ahead, and the animal travelled in a perfect line 100 cm. Now the animal is 100cm from the origin and from the target. But if the animal now overshoots the target by 100cm, they are now 300cm from the origin, and still 100 cm from the target. A similar (more realistic) example happens when the animal does not take an optimal path, but increases their distance from origin without decreasing their distance from the target (e.g., think of a target that is 200cm from the origin / the animal at trial onset, the animal could navigate to form a perfect isosceles triangle, where they are now 200cm from target and origin). More importantly, on different trials the target appears at a random distance between 100cm and 400cm. Thus, if the animal travelled 100cm (imagine in a perfect line to the target), on different trials they could be 0cm from the target (first case), or 300cm from the target (second case). Thus, particularly across trials, there is absolutely no correlation between distance travelled and distance to target, and hence why we can distinguish between these.

We agree with the reviewers that illustrating this point and providing PSTHs (and even more strikingly, rasters) would go a long way to clarifying this issue. We did not perform the exact analysis suggested by the reviewers (because it is not necessarily the case that firing rates vary monotonically with distance, as their suggestion implies), but have added raster plots for example neurons tuned to either the distance from origin or to target. We also show an example neuron that responds to movement stop. This latter neuron, thus, shows a pattern similar to that of Example 2 when sorted from starting location (c.f., Figure 3A third and fifth panel), but critically, when aligned to target location it is evident there is no relation between its firing pattern and the distance to target. Hence, we can differentiate between the distance from origin, to target, and simply responding to moments of starting or stopping movements.

Further, we have modified the main text:

“Beyond the frequency with which we observe neurons tuned to the angle and distance from the origin (i.e., path integration) and to the target (i.e., vector coding of spatial goals), we may expect the distributions of preferred distances and angles to also be informative. Of note, distance/angle from origin and to the target are not the reciprocal of one another given that the target location varies on a trial-by-trial fashion. In other words, the travelled distance and the distance to target may correlate within a trial (but need not, given under- vs. overshooting) but certainly do not across trials (e.g., a distance of, say, 100cm from the origin could corresponding to a whole host of distances from target). In Figure 3A we show rasters of representative neurons tuned to the distance from origin (example neuron 1) and to target (example neuron 2). The neuron tuned to the distance to target (example neurons 2) is not tuned to a particular distance from origin, but does demonstrate a patterned firing rate, discharging at further distances as the animal travels further. The third example (Figure 3A) is tuned to movement stopping, and demonstrates a pattern similar to the neuron tuned to distance to target when plotted as a function of distance from origin (Figure 3A, 3^rd^ vs. 5^th^ panel), but not when visualized as a function of distance to target.

And the figure caption:

“A. Rasters and average firing rate of three example neurons, sorted by their maximal distance from origin and to target. The first example neuron (left) responds at a distance of ~100cm from origin and is not modulated by distance to target. The second example (middle) responds to a close distance to target (~30cm). Arrows at the top of these rasters indicate the preferred distance from origin (example 1) and to target (example 2). We include a third example (tuned to movement stop) as a control, demonstrating that responding to a distance near the target and to stopping behavior are distinguishable.”

3) It would be helpful to provide a few examples of LFP traces and their filtered form along with spike times to appreciate the phase modulations apparent in their statistical modelling results (Figure 2F).

We agree with the reviewers. We have added a new supplement figure (Figure 2 —figure supplement 8), where we show a few example LFP traces, their band-passed version, and the phases. We also show the spikes and how they align to the ongoing phase. Finally, we have also added histograms showing the phase at which a few example neurons fired throughout the course of a recording.

The accompanying figure captions is:

Figure 2 – supplement figure 8. Illustration of spike-LFP phase locking. A. Example trials. For each of four example trials (different sessions as well) we show the raw LFP (top), as well as the band-passed version (transparent) and extracted phase (opaque) in theta (green, second row), α (orange, third row), and β (blue, fourth row) ranges. Spikes are represented by dots, and they are placed on the y-axis according to the phase of the ongoing LFP. That is, across rows spikes occur at the same time along the x-axis, but are at different y-locations. If a neuron is phased-locked to LPF in a given range, spikes should predominantly occur at the same y location (as is seen in these examples for the Β band). B. Example sessions. For 6 example neurons, we show the distribution of phases (x-axis, in radians) at which spikes occurred, throughout the entire session. We show 2 example neurons that were not modulated by LFP phase (1^st^ and 2^nd^ column, uniformly distributed), 2 example neurons that were modulated solely by Β frequency phases (3^rd^ and 4^th^ column), and finally 2 example neurons that were modulated by phases at Theta, Α, and Β frequencies.

4) Tuning strength.From the manuscript it is difficult to judge how representative the different neuronal populations for each area are and to what extent their selectivity differs. The analysis and it variables are quite complex to follow. It would be helpful for the reader to understand how some of them relate to more traditional measures. It is great that the focus on single neurons allows this comparison.What does "tuned" in this context mean in terms of strength and selectivity?How many neurons would pass a minimal response criterion like 10 spikes/s. Would these show stronger tuning or correlations?

We thank the reviewers for this question and consider that in essence they are asking a question about effect sizes. We have added 4 supplementary figures addressing this question (explained below), but must first add the caveat that when examining “raw” firing rates (e.g., response criterion of 10 spikes/s) in a naturalistic task where variables are continuous, dynamic, and correlated, it is not entirely clear what drives the response. This is exactly why we must use a method such as the P-GAM, attempting to factorize variance (i.e., perform credit assignment).

The 4 supplement figures we add (Figure 2 —figure supplement 3, 4, 5, and 6) follow the same format and provide 2 new analyses regarding effect sizes. First, for each sensorimotor (supplement 3), latent (supplement 4), LFP phase (supplement 5) or “other” (supplement 6) variable, we find the neurons that are and are not significantly tuned to that variable. Then, we compute a selectivity index, by computing the difference between the maximum and minimum response of a tuning function (in firing rate space). This gives a more traditional “evoked response”. Then for each population of neurons (significant vs. not) we log transform their evoked response (to render the population Gaussian) and compute Cohen’s d (the distance between means of the distributions normalized by their variance). As it can be observed in only Figure 2 —figure supplement 3, the effect sizes are considerable.

The percentage of neurons showing an increase in firing rate above 10spikes/s for linear velocity, linear acceleration, angular velocity, angular acceleration, the timing of movement onset, offset, and the timing of target presentation are respectively,

For significant units: 10.7%, 34.4%,10.7%, 19.8%, 21.4%, 21.0%, and 3.1%. Non-significant units: 1.5%, 6.0%, 1.1%, 2.3%, 2.2%, 0.5%, and 2.0%.

The second analysis performed (second column) is similar to the first, but contrasting mutual information (a measure of correlation) between evoked responses in the firing rate space and task variables. Findings are conceptually the same as for the first approach.

Associated caption:

“Figure 2 – supplement figure 3. Effect sizes in the firing rate space for neurons deemed to code for sensorimotor variables. Rows are the different sensorimotor variables, in the same order as in Figure 2E and F. The left column is the difference in evoked firing rate between the population of neurons deemed to significantly code for, or not, a given task variable. Namely, for each neuron we compute the difference in firing rate between the peak and the trough of its tuning function. The populations of significant and non-significant neurons are then log-transformed (to render normally distributed) and Cohen’s d is computed (indicated as the title of each subplot). Right column, as for the left column, but while contrasting the mutual information present between firing rates and the given task variable. For reference, Cohen’s d < 0.2 are typically considered weak effects, ~0.5 are considered moderate, and > ~0.8 are considered strong effects.”

Figure 2E: "E. Responses from an example MSTd, 7a, and dlPFC neuron (black), aligned to temporal task variables (e.g., time of movement onset and offset), or binned according to their value in a continuous task variable (e.g., linear velocity)." It would be helpful to give the x-axis for each line for the reader to be able to ascertain what the nature of the scale and the range of variables are over which the firing rate changes are depicted.

We thank the reviewer for this suggestion. An x-axis has been added. The figure is as follows:

Could the authors derive from their data one of the traditional measures used for MST, like a direction tuning index? A direct comparison with previous studies could help understand the nature of the sampled pool, particularly (but not exclusively) for areas when smaller neuronal samples, like MST.

We thank the reviewers for this suggestion. The traditional measure our group (e.g., Chen et al., 2008 in MSTd, and Avila et al., 2019 in 7a) has employed is the “discrimination index”, defined as,DDI=Rmax−RminRmax−Rmin+2SSEN−M Where Rmax and Rmin are the maximum and minimum response from a tuning function, SSE is the sum of squared errors around the mean responses, M is the number of stimulus directions, and N is the total number of observations. In the context of the current naturalistic experiment, we may bin linear and angular velocities and compute tuning functions defining Rmax and Rmin. The number of bins (here we used 15) defines M. To estimate SSE and N we must define trials wherein the full gamut of linear and angular velocities are experienced. To facilitate direct comparison with Chen et al., 2008, we divided our recordings in 80 segments, the mean number of trials (N) in Chen et al., 2008. Lastly, we computed DDI.

Importantly, we plot the mean estimates for MSTd (green), area 7a (blue), and dlPFC (red) from the current dataset in filled triangles, as well as the mean estimates for 7a (from Avila et al., 2019) and MSTd (from Chen et al., 2008) from “traditional experiments”. In general, there is good agreement between studies with MSTd, 7a, and finally dlPFC (in that order) showing the strongest discrimination for optic flow. We must note, however, that there are of course also coarse-grain differences, particularly for MSTd. This is expected given that in contrast to the P-GAM estimate, simply binning data and averaging within these bins does not factorize for the contribution of other experimental factors.

We have included this figure as Figure 2 —figure supplement 7. We have amended the text in the following manners:

Results:

“Similarly, to provide a point of comparison with prior work studying optic flow processing, in Figure 2 —figure supplement 7 we quantify the speed (i.e., liner velocity) and direction (i.e., angular velocity) discrimination index (see Methods and e.g., Chen et al., 2008; Avila et al., 2019) for neurons in MSTd, 7a, and dlPFC.”

Methods:

“To allow for comparison with prior reports studying optic flow processing within the cadre of two-alternative forced-choice tasks, we compute the discrimination index for speed (i.e., linear velocity) and direction (i.e., angular velocity) in MSTd, 7a, and dlPFC. The discrimination index (DDI) was defined as,DDI=Rmax−RminRmax−Rmin+2SSEN−M Where Rmax and Rmin are the maximum and minimum response from a tuning function, SSE is the sum of squared errors around the mean responses, M is the number of stimulus directions, and N is the total number of observations. In the context of the current naturalistic experiment, we may bin linear and angular velocities and compute tuning functions defining Rmax and Rmin. The number of bins (here we used 15 nodes, as defined by the P-GAM) defines M. To estimate SSE and N we must define trials wherein the full gamut of linear and angular velocities are experienced. To facilitate direct comparison with Chen et al., 2008, we divided our recordings in

80 segments, the mean number of trials (N) in Chen et al., 2008. Lastly, we computed DDI according to Equation 5.”

Figure caption:

“Figure 2 – supplement figure 7. Speed and direction discrimination index for neurons in MSTd, 7a, and dlPFC. To allow for direct comparison with prior studies, we compute the discrimination index (see *Methods*) for speed (i.e., linear velocity) and direction (i.e., angular velocity) in MSTd (green), 7a (blue), and dlPFC (red). Full triangles at the top indicate the mean of each population recorded from here (in their corresponding color), while the empty blue triangles show the mean in Avila et al., 2019 (7a recording) and the empty green triangles show the mean in Chen et al., 2008 (MSTd recordings).”

5) Eye movements and visual input.Another issue is the extent that it seems difficult to distinguish the effect of eye position from that of the background stimulus flow patterns, which of course must differ in direction and element size when animals fixate at different locations on the screen. To what extent was this visual input to neurons correlated with "latent variables" like latent distance and angle to target (latent spatial goal)?In order to dissociate the contribution of eye position and task from visual input, do the authors have data on a passive viewing control condition, in which the animal fixates and the visual pattern is played back to animals exactly as if in an active one? How do neural responses compare across the three areas?

These are excellent questions; we address them both together.

In principle there is no correlation between the visual input the animals receive and the latent variables. As it can be observed in Figure 1B (bottom) the animals most often initially move forward and rotate to face the target straight-ahead. Then, after approximately the first 500-1000ms of the trial, they move forward at a constant speed (maximal linear velocity but little to no angular component). Thus, while distance to target is continuously changing (decreasing if the animal is performing the task well), their linear velocity (driving much of the visual stimulation) is held constant.

Nonetheless, the reviewer is correct that the manuscript would be substantially bolstered by providing empirical evidence for the fact that the neural properties we report are in fact related to the animal’s computing taskrelevant metrics and not purely sensory. To address this, we perform exactly what the reviewer suggested. We recorded 2 sessions from area 7a (117 neurons) while the animal first engaged in the task, and then passively viewed the same stimuli replayed (unfortunately recordings in MSTd and dlPFC were not possible at this time). Figure 2 —figure supplement 11 (panel A) shows 7 example trials, showing that stimuli (linear velocity shown, but applies to all task variables) were matched between active and passive conditions. Panel A also shows single trial evoked responses (mean for all simultaneously recorded neurons in the first session, 56 neurons). As it can be observed, there are (single trial) evoked responses when the animal is activity performing the task, but not when passively viewing stimuli. Panel B shows that the mean firing rate (across the entire recording) was unchanged between the active version of the task and passive viewing (dashed line is the diagonal and blue circle is the mean across all neurons). Lastly, Panel C shows summary statistics: the fraction of neurons tuned to different task variables and the fraction of neurons coupled, in active (blue) and passive (black) conditions. All variables except for the phase alignment with LFP bands was significantly blunted in the passive viewing condition. Interestingly, still ~28% of neurons showed tuning to the sensory variables (“latent” and “other” variables average in the passive viewing conditions = 14.8%). Coupling filters were also less prominent in the passive viewing condition (mean ± sem: active = 20.62% ± 0.0047; passive = 15.30% ± 0.0041 ; p = 4.03 x 10^-18^).

The text has been amended in the following manner:

Results:

“In the supplement we demonstrate that this coding was stable (contrasting odd vs. even trials; Figure 2 —figure supplement 10) and task-relevant (Figure 2 —figure supplement 11), in that passive viewing of the same stimuli did not elicit a comparable fraction of neurons tuned to task variables in 7a (passive viewing data in MSTd and dlPFC were unavailable). The fraction of neurons aligned with the phase of LFP in different frequency bands remained stable across passive and active viewing conditions, particularly in the Β band (all frequencies, active vs. passive, p = 0.13; Β band, p = 0.51). Altogether, the encoding pattern across areas may suggest that while dlPFC is critically involved in estimating the relative distance between self and target, 7a may be preferentially involved in the process of path integration, while somewhat unexpectedly, MSTd may play an important role in keeping track of the latent spatial goal.”

And:

“Finer grain unit-to-unit coupling was sparse, and within each area the probability of two units being functionally connected decreased as a function of the distance between neurons (Figure 4 —figure supplement 3A). The overall likelihood of two units being coupled within a given area changed as a function of brain area and was modulated by task engagement (active vs. passive viewing in area 7a; Figure 2 —figure supplement 11C), but not as a function of probe type used (Utah array or linear probe, see Figure 4 —figure supplement 4).”

Figure caption:

“Figure 2 – supplement figure 11. Task engagement drives neural tuning. A. Example trials. To demonstrate that the fraction of neurons tuned to different task variables reported in Figure 2F are driven by task engagement and not purely low-level visual input, in a control experiment (2 sessions) we recorded from area 7a (117 neurons) as a monkey first actively engaged in the task (top), and then passively viewed replayed the exact visual input (bottom). We show 7 example trials, demonstrating that the linear velocity during active and passive trials matched (same for other task variables, not shown). Instead, the population evoked responses (1 trial, average across the entire population of simultaneously recorded neurons) was evident during active but not passive trials. B. Average firing rate. Firing rate (averaged over the entire recording) did not differ between active (x-axis) and passive (y-axis) viewing (blue dots are single cells in 7a, black dot is the mean, dashed black line is identity). C. Fraction of neurons tuned and coupled. The fraction of neurons tuned to different taskvariables, and the fraction of neurons coupled to each other in area 7a, were blunted (but not entirely absent) during passive viewing. The exceptions were variables related to internal neural dynamics, notably the phase locking of spiking activity to LFP phase in Theta, Α, and Β band.”

Further, in the discussion we had previously stated:

“To the best of our knowledge, the striking difference between the posterior parietal node and other areas (here MSTd and dlPFC) vis-à-vis their dependency on the LFP phase has not been previously reported and may have been acutely evident here given the natural timing between sensory inputs, motor outputs, and ongoing neural dynamics that exists within this closed-loop setting”.

This sentence has been removed given the new analysis here demonstrating that even during passive viewing, there was a stronger spike-LFP phase locking in 7a (particularly in theta and α bands) than dlPFC.

Could the authors discuss in the paper how the visual input is (or not) included in the model?

We thank the reviewers for this question. The visual input (in particular the task-relevant feature, being optic flow) was determined by (1) the linear and angular velocity of the animals, and (2) their eye movements. Both of these were included in the P-GAM. However, it is true that the characteristics of each pixel, and how these pixels varied over time, was not included in the model. We consider both of these approaches (ours: in a sense assuming an abstraction of self-velocity; vs. fitting the visual input itself: more a question of representation) to be interesting, yet different. Even complementary. We have modified the discussion in the following manner:

“Performing a “firefly task” in a real environment would also suppose a more complex set of visual inputs (e.g., corners, textures, shadows) that could be leveraged in an expanded P-GAM taking visual features as input (see Parker et al., 2022, for recent work taking this approach).”

6) MSTd and dlPFC coupling.a) As the animals were head-fixed, eye position would compensate in some cases the animals might have moved its head position (for instance to keep track of the target). Both, MSTd and dlPFC encoded eye position. Could the close coupling of MSTd and dlPFC be linked to this element of the task?

We believe that the functional subnetwork established between MSTd and dlPFC reflects the fact that the animals are moving their eyes to keep track of the hidden latent variable. This is shown in Figure 5, demonstrating that the more MSTd and dlPFC were coupled, the more the eyes moved as to keep track of the firefly. In turn, it may be (though this is an empirical question) that if the animals employed another strategy (e.g., moving their heads instead, if freely moving), there would be no (or a reduced) coupling between MSTd and dlPFC. We have amended the discussion in the following manner:

“For instance, virtual and real-world navigation may rely on partially distinct neural codes (Aghajan et al., 2015). Thus, it will be interesting to replicate the current experiment while macaques move freely in a 3D environment (e.g., Mao et al., 2021). This would also allow for independent eye- and head-movements (head was restrained here) and thus we could estimate whether eye movements in the current experiment partially reflected intended head movements (as they seemingly do in rodents; Michaiel et al., 2020, Meyer et al., 2020).”

b) The authors claim that areas MSTd and dlPFC form a functional sub-network together, on the basis of similarity in the fractions of neurons tuned to certain variables, and the distribution of the preferred value of some of these variables. However, the fractions of neurons tuned to the latent variables in MSTd and dlPFC (see Figure 2F) are actually quite different. Perhaps the authors could comment on this.

The reviewer is correct that the fraction of neurons tuned to different variables is not identical in MSTd and dlPFC. However, these areas were in general less tuned to velocity, acceleration, and the timing of different sensorimotor variables than 7a was. MSTd and dlPFC were often tuned to eye position, and these areas were more tuned to the distance to target than 7a. Thus, overall, there were strong similarities between MSTd and dlPFC that did not exist with 7a. Most importantly, we do not make the claim that these areas form a functional subnetwork only because their tuning properties are similar, but because there was a greater likelihood of seeing unit-to-unit coupling between these areas, than between these areas and 7a. We are confident in these results, and in fact initial results regarding population dynamics also supports this claim. Namely, we have seen that the communication subspace (defined as in Semedo et al., 2019, Neuron) is largest between MSTd and dlPFC than between these areas and 7a. And the variable that is most readily decodable from the communication manifold between MSTd and dlPFC is related to eye movements (c.f. Balzani et al., 2022, Arxiv), as would be suggested by the correlation between coupling likelihood and the monkey’s eyes tracking the hidden target.

We consider that it would not be appropriate to include these population-level findings to the current manuscript, as they rely on sophisticated methodology and address a slightly different question: single units vs. population dynamics. Nevertheless, we amend the discussion to explicitly acknowledge that the fraction of units tuned to different variables is not identical in MSTd and dlPFC. The text has been modified in the following manner:

“Similarly, to further corroborate the functional subnetwork between MST and dlPFC it will be interesting to examine population dynamics and the possibility that these areas form a functional “communication subspace” (Semedo et al., 2019), adapted to the naturalistic setting of this task (see Balzani et al., 2022). “

c) When there was stronger MSTd-to-dlPFC coupling and better tracking of the hidden firefly with the eyes (Figure 5B), was the performance of the monkey also better (i.e. more hits)?

We thank the reviewers for this question. Indeed, the better the tracking of the hidden firefly, the better is performance. This is true both when quantified as bias (as we had reported in the original paper and as is reported in Lakshminarasimhan et al., 2020), and when quantified as hit rate (added to the manuscript following this question). Coupling probability (MSTs-to-dlPFC, dlPFC-to-MSTd, and dlPFC-to-dlPFC) does correlate with better gazing toward the firefly. However, this coupling is not predictive of bias (as originally reported in Figure 5 —figure supplement 1), nor of hit rate (as reported now following this question). Together, it appears that (1) coupling within dlPFC or across MSTd and dlPFC correlates with (2) better gazing toward the invisible firefly, and this latter one correlates with (3) better task-performance, but the association between (1) and (3) is not (directly, without (2)) true.

The text has been modified to include the correlations (or lack thereof) with hit rate:

“Further, this relationship also held across sessions, with better target tracking correlating with less bias (slopes closer to 1, see Figure 1C), particularly in the angular domain (r^2^ = 0.43, p = 0.004; radial: r^2^ = 0.26, p = 0.04; Figure 5A), and with an increasing proportion of rewarded trials (r^2^ = 0.24, p = 0.042).”

And:

“As shown in Figure 5 —figure supplement 1, there was no correlation between the unit-to-unit couplings within a session, and either radial or angular biases (7a-7a coupling v. angular bias p = 0.06; all other p > 0.12). There was similarly no correlation between the proportion of rewarded trials in a session and unit-to-unit coupling probability (all p > 0.11, Bonferroni corrected). Overall, therefore, the functional subnetwork between MSTd and dlPFC (Figure 4) seemingly reflects the animals’ strategy in keeping track of the hidden target with their eyes. In turn, the eye movements (but not MSTd-dlPFC coupling directly) aid in successfully navigating to the location of the hidden target.”

7) Sampling of areas.a) Area 7a was exclusively sampled with chronic rather than moveable probes. It has also the largest number of "single units".To what extent are these single units independent?Could a sampling bias in these probes (part of 7a; layers) affect the results, especially when it comes to coupling. Please include in the discussion.

This is an excellent question. We address it by examining the existent recording from Monkey M (dlPFC recordings with linear probes), and by incorporating in the supplementary materials three new recordings (from a new monkey, Monkey B) in 7a, all with linear probes. We fit these sessions to the P-GAM and compute the fraction of neurons coupled within area. Importantly, we only perform this analysis for neurons that were exactly 400 micrometers apart, the minimum distance in Utah probe recordings. Then, we take the array recordings, and compute a new distribution. Namely, we perform 500 iterations in which we first subsample neurons from a randomly selected session to match the number of simultaneously recorded neurons with the linear probes. Then, given that these neurons were at the distance of 400 micrometers, we compute the fraction of neurons coupled. That is, we match the distance (400 micrometers) and number of neurons between array and linear probe recordings. Figure 4 —figure supplement 4 shows, most importantly, that there was no significant effect of probe: the mean of the probe data (dashed black line) fell within the distribution of the array data. Interestingly, it also corroborates Figure 4E, showing that coupling among 7a was more common than among dlPFC.

Figure caption:

“Figure 4 – supplement figure 4. Fraction of neurons coupled in 7a and dlPFC as a function of probe (Utah array or linear probe). We questioned whether the type of probed utilized during recording had an impact on the fraction of units the P-GAM estimated as coupled. To address this question, we examined a new set of recordings in 7a (5 sessions, 32 neurons in monkey B, not reported in the main text), as well as the recordings in dlPFC in monkey M (55 neurons), both of which were conducted with linear probes. For each area, we computed the fraction of neurons coupled during linear probe recordings, given that the distance between these neurons was 400 micrometers, the minimal distance between electrodes in the Utah array recordings. Then, we performed 500 iterations where we randomly subsampled from simultaneously recorded neurons in Utah arrays to match the number of simultaneously recorded neurons with the linear probe. We computed the fraction of neurons coupled within this subsample, again only for neurons at a distance of 400 micrometers. We plot the iterations as a histogram (blue for 7a and red for dlPFC), demonstrating that the fraction of neurons tuned did not significantly depend on type of probe used (7a, mean of linear probe = 0.17, 95%CI of array = [0.10 0.49]; dlPFC, mean of linear probe = 0.10, 95%CI of array = [0.03 0.22]).“

Results:

“Finer grain unit-to-unit coupling was sparse, and within each area the probability of two units being functionally connected decreased as a function of the distance between neurons (Figure 4 —figure supplement 3A). The overall likelihood of two units being coupled within a given area changed as a function of brain area and was modulated by task engagement (active vs. passive viewing in area 7a; Figure 2 —figure supplement 11C), but not as a function of probe type used (Utah array or linear probe, see Figure 4 —figure supplement 4).”

Lastly, we also amend the discussion to make reference to the issues of sampling from a single layer in 7a and the potential of non-independence of single-units (across sessions) during Utah array recordings. The text has been modified as follows:

“The second limitation relates to the (necessarily limited) sampling of neural areas, and the focus on single units as opposed to population dynamics. We report a functional subnetwork between MSTd and dlPFC based on the similarity of their encoding profiles (though they are of course not identical) and the likelihood of encountering unit-to-unit couplings across these areas. But this functional connection must be subserved by structure (e.g., perhaps a third area we did not record from fluctuating with both MSTd and dlPFC). Thus, in ongoing experiments we have trained rodents to perform the “firefly task”. This will allow recording from a wider array of neural areas and cortical layers (most of the recordings reported here being from Utah arrays and hence likely from a single layer and of limited independence).”

b) The number of recorded neurons in the three areas differs greatly: 231 units in MSTd, 823 units in dlPFC, and 3200 units in area 7a. Yet many conclusions in the paper rely on neuronal numbers: the fractions of neurons tuned to certain sensorimotor and latent variables differ between the areas, the variables explaining the firing rates cluster differently in the neurons of the three areas, and both the coarse LFP connectivity and the fine unit-to-unit coupling within areas differ. Especially the clustering results might depend on the number of recorded neurons: the fact that almost all MSTd and dlPFC neurons are categorized as belonging to the same cluster, whereas the area 7a neurons appear in three distinct clusters, could be caused by the much larger number of recorded neurons in area 7a. Also unit-to-unit coupling is more likely to show up in the data with a much larger number of recorded neurons. The data could be corrected for these differences in number of recorded neurons.

We thank the reviewers for this question. Respectfully, we disagree that the fraction of neurons tuned to a particular variable, or the fraction of neurons coupled with others, depends on the number of neurons (these are proportions and thus corrected for the number of neurons). On the hand, we do agree that clustering analyses could be impacted by the relative number of units per area included in the analysis. Thus, we have performed these analyses again, while subsampling from 7a and dlPFC to match the number of neurons present in MSTd.

For the clustering based on whether or not a neuron was tuned to a particular variable (Figure 4A and B), we performed 10k iterations. As highlighted by the reviewers, the striking finding originally reported was that most neurons in MSTd and dlPFC were mixed selective and belonged to a single cluster, while neurons from 7a belonged to a wider variety of clusters. Thus, the metric we compute here is the size of the largest cluster relative to the total number of units. As shown in Figure 4 —figure supplement 1 (panel A, means and 95%CI shown) even when matching the number of units across areas, ~70% of units in MSTd and dlPFC come from the most populous cluster, while only ~35% do so in 7a. This results thus confirm the original finding.

Regarding the clustering based on tuning function shapes and UMAP projections (Figure 4C), the key metric originally reported was that the centroid of the MSTd UMAP projection was 6.49 times closer to the centroid of dlPFC than to the centroid of 7a. We computed this metric again, while performing 100 iterations. Figure 4 —figure supplement 1 (Panel B) we depict the ratio of MSTd-to-dlPFC UMAP distance to MSTd-to-7a distance, for all iterations. As it can be observed, the finding holds, with the boundaries of the 95%CI being 4.07 and 7.33.

The results and figure caption have been amended:

Results:

“Other cluster types existed, for instance composed of neurons selectively tuned to the ongoing phase in LFP bands but no other task variable (Figure 4A and B, Cluster 4), or driven also by motor onset and offset (Figure 4A and B, Cluster 5). These remaining clusters were, however, less common (~1-5%). This analysis was conducted with the full dataset (4254 neurons in total), yet in the supplement (Figure 4 —figure supplement 1A) we confirm that the results are unchanged when subsampling from areas with more neurons (7a and dlPFC) to match the number present in MSTd (231 neurons). Together, this pattern of clustering results based on whether neurons were tuned to different task variables demonstrated a surprising commonality between MSTd and dlPFC, which are in turn different from area 7a.”

And:

“Notably, however, the centroid of MSTd was 6.49 times closer to the centroid of dlPFC than area 7a (Figure 4C, top row. Note that Becht et al., 2018, have shown UMAP to conserve global structure and thus allows for a meaningful interpretation of distances). This finding also holds when subsampling from 7a and dlPFC to match the number of units present in MSTd (100 iterations, MSTd-dlPFC distance was 5.56 times closer than MSTd7a, 95%CI = [4.07, 7.33]; Figure 4 —figure supplement 1B).”

Figure caption:

“Figure 4 – supplement figure 1. Clustering results, subsampling from neurons in dlPFC and 7a to match the number of units recorded from in MSTd. A. We performed spectral (Jaccard) clustering of neurons based on their 1 x 17 vector of Booleans, indicating whether they were tuned or not to particular task-variables (Figure 4A). Here, we perform this operation 10000 times, while randomly selecting (without replacement) 231 neurons from 7a and dlPFC. Thus, the full matrix clustered was 693 (231 x 3 brain areas) x 17 (task variables). Given that on each run clusters are assigned an arbitrary cluster number, for each run we compute the ratio of the largest cluster size to the total number of units per area (231). Namely, in the main text we report that MSTd and dlPFC are predominantly represented by 1 mixed-selective cluster, while 7a is represented in 3 approximately equal sized clusters. The results here concord with those in the main text, demonstrating that approximately 70% of neurons in MSTd and dlPFC belong to a single cluster, while approximately 35% of neurons belong to the largest cluster in area 7a. Circles are the mean across 10000 iteration, error bars are 95%CI. B. 100 iterations of UMAP while randomly subsampling from 7a and dlPFC to match the number of units in MSTd. On each run, we compute the distance in UMAP space between MSTd and dlPFC, and between MSTd and 7a. Then we compute their ratio (MSTd-to-dlPFC/MSTd-to-7a, thus >1 indicating closer MSTd-to-dlPFC distances). The figure shows the full distribution of ratios, concurring with the main text that MSTd and dlPFC are approximately 6 times closer in UMAP space, than MSTd and 7a are. “

8) Lateralisation.To what extent played the lateralization of the recording and task a role for neuronal response? This applies relative to brain hemisphere, body and eye position? Where in each monkey did the recordings take place? Which hand(s) did each monkeys use for the choice stick?How was the lateralisation included in the model?Please comment with regards to responses in MST, 7A, and dPFC and add information to the manuscript.Specifically, it is unclear from Suppl Figure 1 whether within a particular monkey, some recording sites were interhemispheric, or whether within one monkey, all recordings were done in the same hemisphere. This of course has significant consequences for the effects of ongoing LFP and unit-to-unit coupling.

We thank the reviewers for highlighting that Figure 1 —figure supplement 1 did not provide enough information as to lateralization. We have significantly expanded this figure caption to add the information requested by the reviewers. The figure caption reads:

“Location of acute recordings are indicated by spheres, color coded per animal (Monkey S, orange; Monkey Q, purple; Monkey M in green). Location of Utah arrays are indicated by squares. The pictures of the Utah arrays are framed in the color corresponding to the monkey. In turn, Monkey S had recordings performed from Utah arrays in dlPFC and area 7a on the left hemisphere, and from a linear probe in MSTd on the right hemisphere. Monkey Q had recordings performed from a Utah array in 7a on the left hemisphere and from a linear probe in MSTd on the right hemisphere. Monkey M had recordings performed from a linear probe in dlPFC on the right hemisphere. In total, therefore, each area was sampled twice. All 7a recordings were on the left hemisphere and all MSTd recordings were on the right hemisphere (note, given that MSTd is directly ventral to 7a, see Figure 2A, these cannot be recorded from the same hemisphere if the former area is implanted with an array). From the 823 neurons recorded in dlPFC, 55 were recorded on the right hemisphere in Monkey M. All inter-area coupling analysis (which requires simultaneous recordings) were based on left-hemisphere recordings in 7a and dlPFC, and from right-hemisphere recordings in MSTd. All monkeys were right-handed and used this hand to manipulate the joystick. AS, arcuate sulcus; IPS, intraparietal sulcus; PS, principal sulcus; STS, superior temporal sulcus; LF, lateral fissure.”

In short, all animals were right-handed. All recordings in area 7a were on the left-hemisphere, while all recordings in MSTd were on the right hemisphere. It is impossible to record from these two areas simultaneously in the same hemisphere, if recording high-density with a Utah array in 7a (the area on the surface). The vast majority of neural recordings in dlPFC (93%) were conducted via a Utah array on the left hemisphere. All data analysis requiring simultaneous recordings (i.e., inter-area coupling, inter-area LFP phase coupling) were conducted with MSTd data on the right hemisphere, and 7a and dlPFC on the left hemisphere. Thus, 7a and dlPFC were on equal footing with regard MSTd, in that they were both inter-hemispheric.

9) Data fed into the P-GAM model.a) The P-GAM model is a great analysis tool for these kinds of data. However, the variables that the authors put into it are conceptually very different from each other. There are purely external task variables such as target onset and offset, latent variables such as distance to target that require knowledge of one's own position in space, and purely internal brain dynamics variables such as coupling to the LFP in another area. In that light, the finding of 'many variables contributing to the responses' is not surprising; all neurons in the brain are probably influenced both by external variables and internal brain dynamics. Maybe the authors could comment on the different nature of their variables and how that impacts their results.

We agree with the reviewer that neurons are very likely influenced by both external task variables and internal brain dynamics. In fact, we consider that the inclusion of both these types of variables to be a strong asset of the current manuscript. We have amended the text to more explicitly mention the difference between these types of variables. The text reads:

“In addition to continuous sensorimotor (e.g., linear and angular velocity and acceleration) and latent variables (e.g., distance from origin and to target, Figure 1A), as well as discrete task events (e.g., time of target onset, as well as movement onset and offset), we included elements of brain dynamics in the encoding model. These internal dynamics are most often not considered in accounting for task-relevant neural responses, yet they fundamentally shape spiking activity. These latter variables included the phase of LFP in different frequency bands (theta: 4-8 Hz; α: 8-12 Hz; β: 12-30 Hz), and causal unit-to-unit coupling filters within (i.e., spikehistory, 36 ms wide temporal filter) and between units, both within (36 ms wide temporal filter) and across cortical areas (600 ms wide temporal filters, Figure 2C, see *Methods*).”

b) Given that the sensorimotor and latent variables going into the G-PAM model are so crucial for the story, could you make a figure where you visualize them? This could maybe be added to Figure 1A. Also, 'radial bias' and 'angular bias' (in Fig1D) could be visualized here.

We thank the reviewers for the suggestion and agree that depicting these time-courses could be helpful to readers. We have modified the figure.

c) Quantification of electrophysiological activity processing that is fed into the P-GAM model is not entirely clear.More details about the preprocessing of these data are required, for example, are the SUA baselined using pre-stimulus presentation activity? Are the LFP baselined as well? And how similar are the pooled responses within each area and across? This would allow the reader to spot possible problems when computing further neuronal properties, that could bias the main paper result:An example is the tuning of the neurons to the phase of ongoing oscillation (Β, Α, Theta). There are a number of papers attempting to optimize methods to measure spike field coherency, e.g. the PPC pairwise phase consistency (Vinck et al., 2010). This method gives an estimation independent of spike count and LFP amplitudes (both parameter vary of course widely across time, tasks, subjects, areas…).Here, it seems these two parameters are not considered and could lead to artefacts in the coupling results presented. The authors use temporal correlations to approximate coupling between spike/spike, and spike/LFP-phase. Correlation methods can potentially lead to artefacts and overestimations of coupling strength.In their methods, the author state to 'bin spiking activity across 8ms window' prior to feeding this activity to the P-GAM. It means that 1 spike corresponds to an averaged 8ms time window. If you now try to calculate the dependency of this single spike to a specific phase of a β (30Hz) , the α (12Hz) and theta (4Hz) oscillation, it means that the chance level of assigning the binned spike to a particular phase differs considerably. Therefore, the statistical power of this analysis would decrease for higher frequency. It seems that the authors do not apply any correction.

We thank the reviewers for this question. We have added information regarding the pre-processing of neural data in the following manner:

“As such, inputs to the P-GAM were of three types. First, the spike counts of the unit to be modelled, at a 6ms resolution (i.e., the number of spikes within 6ms windows, no baseline correction). Second, the continuous, discrete, and neural co-variates, which were also sampled at a 6ms resolution. The last input type were a set of 15 “knots” per co-variate, defining the nodes of eventual tuning functions. The location of knots were defined as to (i) cover the range of a given input variable from the second to the 98^th^ percentile with (ii) equi-probable knots (each knot covering the same probability mass). See

https://github.com/BalzaniEdoardo/PGAM/blob/master/PGAM%20Tutorial.ipynb for a comprehensive tutorial.“

Regarding the spike-LFP phase coupling, we thank the reviewers for highlighting the pairwise phase consistency (PPC0) measure. We have applied this measure and report the findings in Figure 2 —figure supplement 9.

As it can be observed, the main finding that neurons in area 7a are considerably more phase locked to LFPs (particularly in the Β range, but also in Α) holds. We have added Figure 2 —figure supplement 9 and the following text:

Results:

“Neurons in area 7a showed a strong dependency to the phase of ongoing LFP fluctuations (see Figure 2 —figure supplement 8 for further illustrations of this effect as quantified by the P-GAM, and Figure 2 —figure supplement 9 for corroborative evidence by pairwise phase consistency, Vinck et al., 2010), a fact that was less observed in dlPFC or MSTd (all p < 7.1 x 10^-8^, all d > 1.02) “

Figure caption:

“Figure 2 – supplement figure 9. Pairwise phase consistency. The findings related to spike-LFP phase coupling reported in the main text are based on the P-GAM, as are the rest tuning properties reported. However, detecting a correlation between when spikes occur and the phase of LFPs may be biased by a number of factors, for example the firing rate of neurons or the amplitude of LFP oscillations. In turn, we corroborated the spikeLFP phase coupling results by computing the pairwise phase consistency (PPC0), a bias-free estimator (see Vinck et al., 2010 for detail). This analysis confirmed the findings from the main text in demonstrating greater spike-LFP phase coupling in area 7a, particularly within the Β and Α ranges. Mean PPC across all neurons and sessions are reported separately by brain region and frequency range. Error bars are S.E.M.”